# Cold temperature induces a TRPM8-independent calcium release from the endoplasmic reticulum in human platelets

Anastasiia Stratiievska[1], Olga Filippova[1], Tahsin Özpolat[1], Daire Byrne[1], S. Lawrence Bailey[1], Aastha Chauhan[1], Molly Y. Mollica[1,2,3], Jeff Harris[1], Kali Esancy[4], Junmei Chen[1], Ajay K. Dhaka[4], Nathan J. Sniadecki[5,6], José A. López[1,2], Moritz Stolla[1,2,6]*

1 Bloodworks Research Institute, Seattle, WA, United States of America, 2 Department of Medicine, Division of Hematology, School of Medicine, University of Washington, Seattle, WA, United States of America, 3 Department of Mechanical Engineering, University of Maryland, Baltimore County, Baltimore, MD, United States of America, 4 Department of Biological Structure, University of Washington, Seattle, WA, United States of America, 5 Department of Laboratory Medicine and Pathology, University of Washington, Seattle, WA, United States of America, 6 Department of Mechanical Engineering, Bioengineering, University of Washington, Seattle, WA, United States of America

* mstolla@bloodworksnw.org

**Data Availability Statement:** All relevant data are within the paper and its Supporting Information files.

## Abstract

The detection of temperature by the human sensory system is life-preserving and highly evolutionarily conserved. Platelets are sensitive to temperature changes and are activated by a decrease in temperature, akin to sensory neurons. However, the molecular mechanism of this temperature-sensing ability is unknown. Yet, platelet activation by temperature could contribute to numerous clinical sequelae, most importantly to reduced quality of *ex vivo*-stored platelets for transfusion. In this multidisciplinary study, we present evidence for the expression of the temperature-sensitive ion channel transient receptor potential cation channel subfamily member 8 (TRPM8) in human platelets and precursor cells. We found the TRPM8 mRNA and protein in MEG-01 cells and platelets. Inhibition of TRPM8 prevented temperature-induced platelet activation and shape change. However, chemical agonists of TRPM8 did not seem to have an acute effect on platelets. When exposing platelets to below-normal body temperature, we detected a cytosolic calcium increase which was independent of TRPM8 but was completely dependent on the calcium release from the endoplasmic reticulum. Because of the high interindividual variability of TRPM8 expression, a population-based approach should be the focus of future studies. Our study suggests that the cold response of platelets is complex and TRPM8 appears to play a role in early temperature-induced activation of platelets, while other mechanisms likely contribute to later stages of temperature-mediated platelet response.

## Introduction

The detection of temperature change by nerve receptors in human tissues is a life-preserving function that is highly evolutionarily conserved. Non-neural cells also sense cold, a prime

**Funding:** Funding sources from NIH: HL147462 and HL007093 to MYM; S10 OD016240 to Keck center; 5R01NS115747-03 to AKD; HL145262 and HL149734 to NJS; R35HL145262 JAL; 1R01HL153072-01 to MS; and institutional funds from the Bloodworks Northwest. The funders had no role in study design, data collection and analysis, decision to publish, or preparation of the manuscript.

**Competing interests:** M.S. received research funding from Terumo BCT and Cerus Corp. All other authors have no COI to declare. This does not alter our adherence to PLOS ONE policies on sharing data and materials.

example being blood platelets. Platelets are anucleate blood cells, derived from megakaryocytes in the bone marrow, critical for vascular integrity and hemostasis [1]. Platelets are activated below body temperature which leads to low-level integrin activation and shape change induced by cytoskeletal rearrangement [2, 3]. The pathophysiologic mechanism of this response is unknown, perhaps these changes prime platelets for activation at more injury-prone extremities, where body temperatures are cooler than at the core of the body, to minimize blood loss in conjunction with vasoconstriction. The platelet activation by temperature could contribute to several clinical observations. For example, myocardial infarctions occur more frequently during colder winter months [4]. The reasons for this increase are likely multifactorial, but platelet inhibition with acetylsalicylic acid (aspirin) is protective, suggesting platelet activation is of major importance [5, 6]. Platelets for transfusion are stored at room temperature resulting in a progressive functional and morphological decline [7]. Recent clinical trials cast doubt on the safety and efficacy of RT-stored platelets [8]. Further reducing the storage temperature to 4˚C accelerated clearance to an extent that they were no longer considered useful for patients who require long circulating platelets [9]. Therefore, blood banks worldwide store platelets at room temperature to maximize post-transfusion circulation time. Room temperature storage is onerous, expensive, and leads to a maximum shelf life of only seven days due to the risk of bacterial growth. Patients after cardiac arrest often undergo therapeutic hypothermia and can develop unexplained thrombocytopenia in this process [10–13]. Lastly, the physiologic response of platelets to cold temperatures also has implications for the interpretation of basic and translational research studies. To isolate platelets for research experiments, whole blood is routinely cooled to room temperature after phlebotomy to allow for further processing. This exposure to temperature ranges below 37˚C may explain some of the differences observed between *in vivo* and *in vitro* platelet testing and could have implications for platelet function studies that are based solely on room temperature data. Taken together, understanding, and mitigating the platelet temperature response is critical for patients from several different medical fields [14].

A variety of possible mechanisms for cold-induced activation of platelets have been proposed: lipid phase transition, calcium leakage from intracellular stores, and reduced calcium exchanger activity have shown some involvement, but the exact molecular mechanism is unknown [15, 16]. Like neurons, platelets express a wide variety of ion channels on their surface that are involved in rapid intracellular signaling in response to different physical and chemical extracellular stimuli [17]. Ion channels are transmembrane receptors with an ion-permeable pore that opens to allow ion flux in response to stimuli. One ion channel of interest is transient receptor potential cation channel subfamily member 8 (TRPM8), which is activated by temperatures below 26˚C and cooling compounds such as menthol [18, 19]. The temperature activation profile of TRPM8 channels makes it a good candidate for a molecular entity behind the platelet cold response.

In this study, we sought to investigate whether TRPM8 is involved in the acute cold response in platelets. We found a moderate level of TRPM8 expression and function in the human megakaryocyte cell line. A small population of platelets was positive for TRPM8 staining. Cold-induced platelet shape change and αIIbβ3 integrin activation were partially TRPM8-dependent, while we were unable to detect any TRPM8-dependent platelet activation in aggregation and calcium influx assays. Finally, temperature-induced calcium influx in platelets was not via TRPM8, but fully dependent on the calcium release from the endoplasmic reticulum (ER). Taken together, for the first time we show TRPM8 expression in platelets, but its physiological role remains incompletely understood. In addition, the molecular entity responsible for cold temperature-induced calcium release from the ER in platelets needs to be further investigated.

## Results

### TRPM8 mRNA in MEG-01 cells

Because platelets are generated by megakaryocytes, we first evaluated TRPM8 expression in MEG01 cells, a megakaryocytic cell line. We amplified TRPM8 transcripts by PCR using mRNA isolated from cultured MEG-01 cells using primers designed for human TRPM8 sequences (Fig 1A). The effectiveness of primers was confirmed on purified human TRPM8 cDNA (Fig 1A). We observed clear bands of the expected sizes representing TRPM8 mRNA from both MEG-01 passages tested (Fig 1A, MEG-01 P1, P2). These findings are consistent with the expression of TRPM8 in primary human megakaryocytes (S1 Fig) [20, 21]. We conclude that there is TRPM8 mRNA expression in megakaryocytes.

### TRPM8 protein in MEG-01 cells

We tested for TRPM8 protein expression in MEG-01 cells by Western blot. We observed a strong band at the expected size of ~130 kDa, corresponding to full-length TRPM8. Interestingly, we also observed two smaller bands, at about 60 and 45 kDa, suggesting the expression of shorter TRPM8 isoforms or protein degradation (Fig 1B). Next, we visualized TRPM8 on the surface of the MEG-01 cells using immunostaining. To test for TRPM8 surface expression, MEG-01 cells were left unpermeabilized and stained with an αIIb integrin antibody (CD41, Fig 1C, magenta), a Hoechst nuclear stain (Fig 1C, blue), and an antibody raised against the peptide corresponding to the extracellular epitope of the TRPM8 (Fig 1C, yellow). During maturation, megakaryocytes form a complex system of membrane invaginations called the demarcation membrane system (Battinelli et al., 2001). We observed that the membrane-localized αIIb (CD41) staining was increased in areas suggestive of multiple layers of the plasma membrane, possibly the demarcation system (Fig 1C). A similar pattern of expression was observed with TRPM8 staining in MEG-01 cells suggesting membrane expression. Considering that both αIIbβ3 integrin and TRPM8 are transmembrane proteins, these data suggest that TRPM8 is expressed on the surface membrane of MEG-01 cells.

### TRPM8 function in MEG-01 cells

To test for TRPM8 function, we performed calcium imaging on Fluo4-loaded MEG-01 cells and applied TRPM8 agonists in a controlled flow chamber. Fig 1D, and 1DE show normalized calcium responses to the application of either vehicle, TRPM8 agonists, or the platelet agonist adenosine diphosphate (ADP). We observed that the application of the agonists elicited small responses in a subpopulation of cells (Fig 1D–1F). Because not all MEG-01 were responsive to TRPM8 agonists, we separated the cells into two groups based on their responses (Fig 1F, positive or negative). A fraction of MEG-01 cells showed a small response to vehicle ETOH 0.1% application (35% cells, Fig 1F). Considering this baseline responsiveness to the vehicle, we only compared the delta calcium increase from cells showing positive responses (Fig 1G). In the positive (i.e., agonist responsive) fraction of MEG-01 cells, the vehicle led to the average $0.056 \pm 0.004$ (Mean ± SEM) calcium increase in $\Delta F/(F_{max}-F_0)$. The TRPM8 agonist menthol did not significantly increase calcium (Fig 1D and 1G, blue), and the positivity rate of menthol responses was not different from the vehicle (Fig 1F, 33.1% and 35.3% respectively). In addition, we used WS-12– a more specific TRPM8 agonist structurally similar to menthol [22], as well as icilin–a super-cooling agent, which is structurally distinct from menthol [19]. In contrast to menthol, WS-12 and icilin, led to a significantly higher calcium influx than vehicle (Fig 1D and 1G, Mean ± SEM = $0.12 \pm 0.02$ and $0.08 \pm 0.01$, $p$-value = 0.005 and 0.03, respectively). WS-12 and icilin-responsive cell populations were larger than the vehicle's positivity rate (Fig

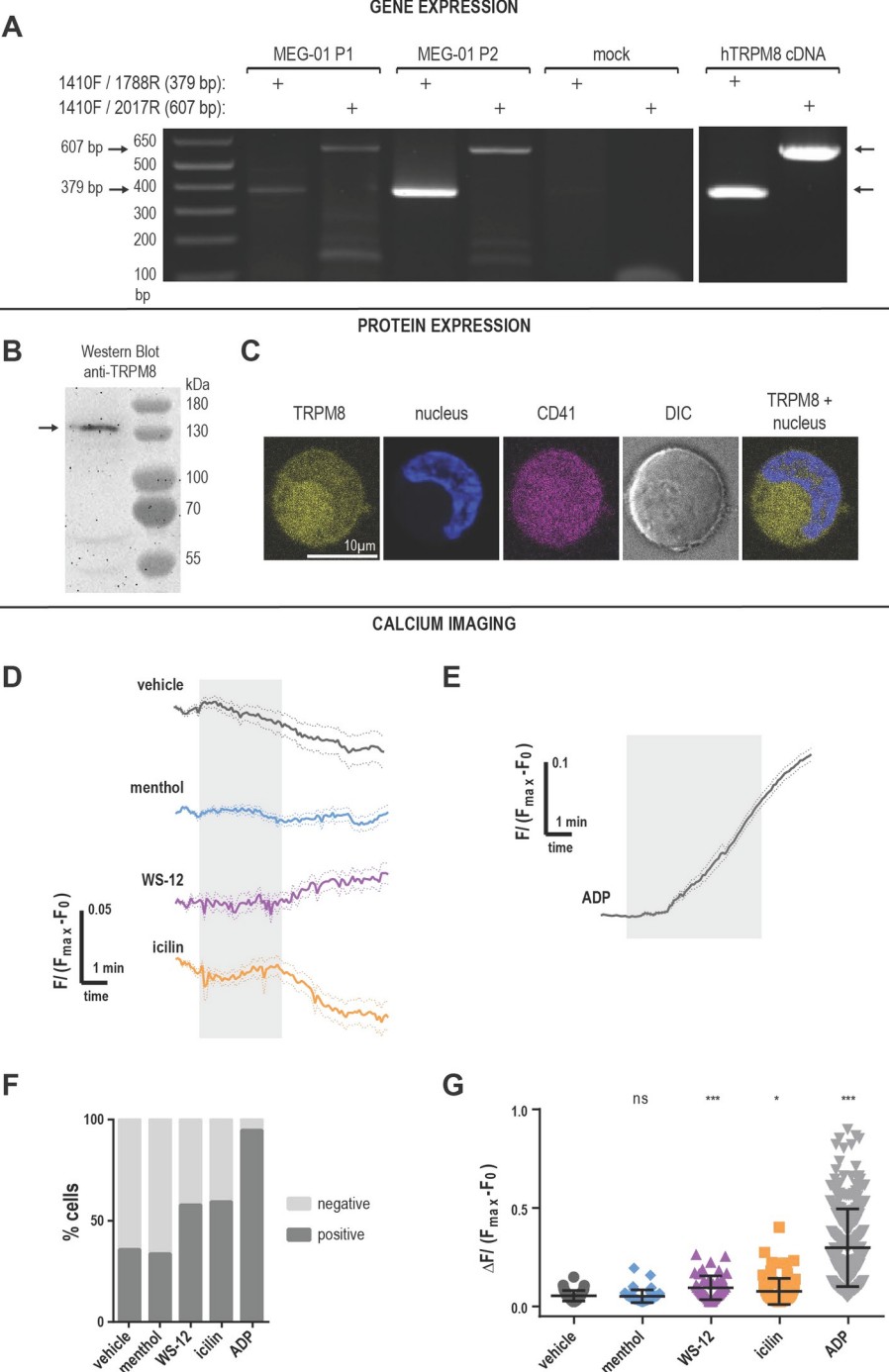

**Fig 1. TRPM8 is expressed in MEG-01 cells. A**. Agarose gel electrophoresis of PCR products from MEG-01 cells, using two sets of TRPM8 primers. Arrows indicate the size of the expected amplicons: 379 bp for the 1410F/1788R primer set, and 607 bp for the 1410F/2017R set. P1 and P2 indicate unique MEG-01 passages. The preparation was tested for contamination by using mock samples including the primers and the PCR reaction kit without cDNA. **B**. Anti-TRPM8 western blot of MEG-01 cell lysate using the primary anti-TRPM8 antibody (#NBP1-97311, raised against the peptide corresponding to the intracellular epitope) and the secondary goat anti-rabbit HRP-conjugated antibody. **C**. Confocal images of MEG-01 immunostaining with anti-TRPM8 (yellow; primary antibody raised against the peptide corresponding to the extracellular epitope + goat anti-rabbit secondary), Hoechst nuclear staining (blue), integrin IIb/IIIa (CD41, magenta), DIC (gray) and TRPM8 + nucleus overlay. The scale bar is 10 μm. **(D-G)** Calcium imaging of Fluo4-AM loaded MEG-01 cells. **D**. TRPM8-dependent calcium influx in MEG-01 cells. Average responses to TRPM8 agonists over time from responsive MEG-01 cells. Vehicle EtOH 0.1% (gray), menthol 500 μM (blue), WS-

12 4 μM (magenta), icilin 100 μM (orange). Fluo4-AM fluorescence (F), baseline ($F_0$) corrected and normalized to $F_{max}$ obtained after ionophore 23187 application. Thick lines indicate average and dotted lines indicate SEM. The shaded area indicates the duration of agonist application. The delay in the WS-12 response might be due to the delay in perfusion combined with the low working concentration. **E.** Average response over time to ADP 50 μM. **F.** Percent MEG-01 cells showing positive (dark gray) and negative (light gray) responses to agonists. **G**. Change in calcium levels from individual "positive" cells in response to agonists: vehicle EtOH 0.1% (dark gray), menthol 500μM (blue), WS-12 4 μM (magenta), icilin 100 μM (orange), and ADP 50 μM (light gray). Error bars indicate mean ± SEM. Symbols above scatter graphs indicate p-values from unpaired Student's t-test comparison with vehicle (ns–p > 0.05, *–p < 0.05, **– p < 0.01, ***–p < 0.005).

1F, 57.3% and 59%, respectively). Taken together, our data suggest that TRPM8 is functional at low levels in a population of platelet precursor MEG-01 cells.

## TRPM8 protein in human platelets

We examined purified human platelets for the presence of TRPM8 messenger RNA. We found evidence for TRPM8 mRNA, but were unable to rule out leukocytes as source of TRPM8 mRNA (S2 Fig). Therefore, for the detection of TRPM8 protein expression in platelets, we used an antibody specifically recognizing an extracellular epitope of TRPM8. Using Western blot, we detected several bands specific to the TRPM8 epitope, but there was a very low expression of the full-length TRPM8 (S3 Fig). Next, we examined TRPM8 expression in human platelets by flow cytometry (see Fig 2A–2C for gating strategy). We observed that platelets from human platelet-rich plasma (PRP) were on average 11.2% ± 2.3 (Mean ± SEM, n = 7) TRPM8-positive (Fig 2D).

To visualize TRPM8 and gather more information about its specific localization in platelets, we used imaging flow cytometry. Platelets were identified by αIIb integrin staining (CD41). As shown in Fig 2E, platelets positive for TRPM8 show a punctate staining pattern, which tends to localize to the periphery of the platelet. In some instances, the TRPM8-positive foci were localized outside of the platelet outline, seemingly attached to the platelet with a thin undistinguishable protrusion (Fig 2E, middle row). Interestingly, within the TRPM8-positive platelet population, there was a significantly lower percentage of spheroid cell shape and a significantly higher percentage of discoid cells (Fig 2F). Thus, a small platelet population is positive for the surface expression of TRPM8.

We also sought to visualize TRPM8 channels on platelet footprints attached to glass surface using immunostaining and confocal microscopy. Like our findings by imaging flow cytometry, we observed anti-TRPM8 staining in a punctate pattern (Fig 2G and 2H left). Samples stained with only the fluorescent secondary antibody were not distinguishable from the background (Fig 2G and 2H right). TRPM8 foci tended to localize to the periphery of the platelet footprint. To find the platelet outline, we stained all samples with the lipophilic plasma membrane fluorescent dye R18 (Fig 2G, white outline). We observed that 68.4% ± 5.0 (Mean ± SEM, n = 3) of all platelet membranes were positive for TRPM8 staining. To evaluate the possibility that TRPM8-containing microparticles derived from CD45-positive white blood cells (WBC-MP) are the source of platelet TRPM8, we incubated WBC-MP with platelets. Indeed, incubation of platelets with WBC-MP led to a significant TRPM8 increase compared to platelets alone or WBC-MP alone (20661 MFI ± 1415 versus 3732 MFI ± 360 and 4754 MFI ± 1794, respectively) (S4A Fig). However, the TRPM8 expression on platelets was not dependent on WBC-MP as the TRPM8 events on platelets without WBC-MP co-incubation did not necessarily overlap with CD45 expression (S4B and S4C Fig). In conclusion, we show that a subpopulation of human platelets expresses TRPM8.

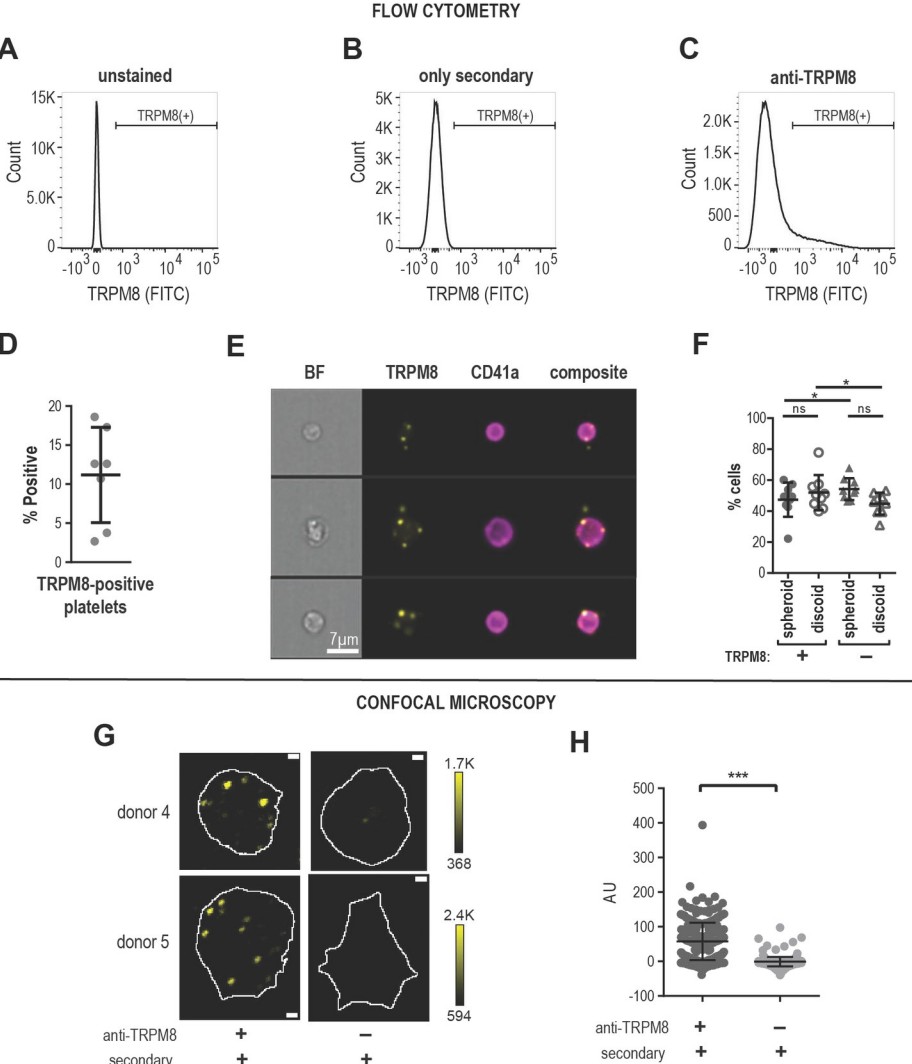

**Fig 2. TRPM8 receptor expression in human platelets. (A-F)** Flow cytometry analysis of human platelets in plasma. Cell count vs fluorescence intensity histograms of unstained **(A)**, stained with only secondary antibody **(B)** and anti-TRPM8 primary with secondary FITC-conjugated antibody staining **(C)**. TRPM8 (+) platelets defined based on secondary control samples, brackets indicate the TRPM8-positive platelets. 100,000 events were measured for each sample. **D**. Percent of TRPM8 positive platelets from several different healthy donors (n = 7). **E**. Representative images of random CD41 and TRPM8-positive platelets from one healthy donor (out of five) by imaging flow cytometry. 20,000 events were measured for each sample. The scale bar is 7 μm. **F**. Percentage of spheroid and discoid cells within TRPM8-positive (+) and -negative (-) platelet populations. Platelets were assigned spheroid or discoid shapes as described in (Özpolat et al., 2023). **(G-H)** Confocal microscopy of anti-TRPM8 immunostaining of washed platelets on poly-lysine coated glass coverslips. **G**. Representative images of platelets from two healthy donors stained with primary anti-TRPM8 and secondary antibodies (left) or secondary-only control (right). The white outline indicates the footprint of the platelet as identified by the R18 membrane stain. The scale bar is 1 μm. **H**. Background subtracted average pixel intensity (AU) of platelets from two healthy donors (stained with primary anti-TRPM8 and secondary antibodies (268 cells) or secondary only control (142 cells). Error bars indicate Mean ± SEM. Asterisks indicate the p-value <0.001 from the unpaired t-test with Welch's correction for unequal distribution comparing background subtracted ROI intensities.

## TRPM8 function in human platelets

Cooling platelets from the body temperature to room temperature activates platelets [23]. Nevertheless, it is standard practice in platelet biology research to conduct experiments at room temperature. Given the evidence of TRPM8 surface expression in human platelets, we hypothesized that temperature change at the time of collection influenced the level of platelet activation in a TRPM8-dependent manner.

First, we compared levels of platelet activation between the samples collected from the same donor at varying temperatures in the presence and absence of the specific TRPM8 inhibitor, PF 05105679 [24, 25]. Platelet αIIbβ3 integrin activation can be measured by the activation-specific PAC-1 antibody [26], which we used to determine the level of platelet activation in whole blood. We observed significantly more αIIbβ3 activation in platelets collected into room temperature (22˚C) tubes (Mean ± SEM = 13.4 ± 3.2% PAC-1-positive cells, Fig 3A) than in blood collected into pre-warmed tubes (37˚C) (Mean ± SEM = 1.4 ± 0.4% PAC-1-positive cells Fig 3A).

We hypothesized that inhibition of the TRPM8 channel during the collection at room temperature would lead to decreased platelet activation. It has been previously reported that the temperature threshold of TRPM8 activation depends on several intra- and extra-cellular factors [27–29]. A decrease in temperature from 37˚C to 22˚C could lead to the opening of the TRPM8 channel and therefore, higher intracellular calcium levels that activated platelets. First, we found that the addition of PF 05105679 into the pre-warmed to 37˚C syringe during blood collection, did not significantly affect the levels of integrin-activated cells, compared to the vehicle (Fig 3A, Mean ± SEM = 1.4 ± 0.3%, $p$ = 0.85). In contrast, when blood was collected at 22˚C, PF 05105679 treatment reduced the percentage of platelets that were PAC-1 -positive compared to vehicle, to the degree that approached statistical significance (Fig 3A, Mean ± SEM = 7.9 ± 1.3%, $p$ = 0.06). The inhibition by PF 05105679 at 22˚C failed to reach the baseline, suggesting some level of TRPM8-independent activation by 22˚C temperature during collection. In addition, the TRPM8 inhibitor did not affect the activation of platelets exposed to 4˚C (Fig 3A, green). These data implicate TRPM8 in αIIbβ3 activation in platelets exposed to room temperature.

We also analyzed microaggregates by imaging flow cytometry. Microaggregates are small clusters of platelets, a feature of the pro-aggregatory platelet status, and used as a tool to evaluate the antithrombotic effects of compounds [30, 31]. In a recent study, the presence of microaggregates was responsible for the wastage of a significant proportion of cold-stored platelet units [32]. To evaluate the effects of temperature and TRPM8 inhibition on the microaggregate content in our samples, we collected blood into pre-warmed tubes (37˚C) containing either vehicle or PF 05105679. After incubation for 5 min, the samples were cooled down to 22˚C or 4˚C for 15 min. Fig 3B shows the percent microaggregates under these conditions. There were no significant differences in microaggregate formation at either temperature (Fig 3B), nor any significant effect of PF 05105679 (Fig 3B, open symbols). This suggests that the inhibition of TRPM8 during cooling does not reduce the propensity of platelets to form aggregates.

## The temperature-induced platelet shape change is TRPM8-dependent

Platelets change morphology, from discoid to spheroid, upon cold exposure [15, 33]. We recently developed an unbiased screening protocol for differentiating between the two morphological states using imaging flow cytometry [34]. The 2D images of 3D discs and spheres are accurately described as fusiform (spindle-shaped) and circular shapes, but, for simplicity, we will refer to these as discoid and spheroid.

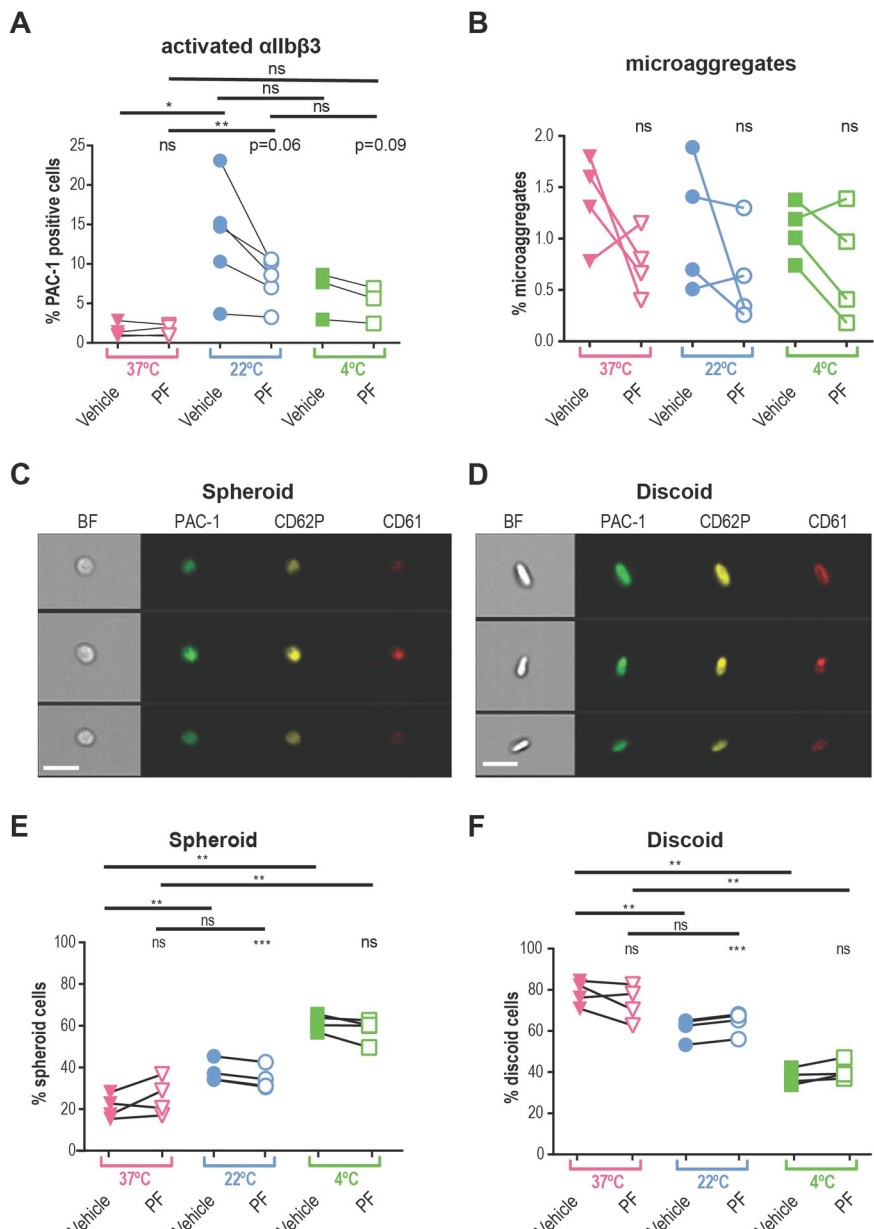

**Fig 3. Platelets are activated by room temperature in TRPM8 dependent manner. A.** Expression of the active form of integrin αIIbβ3 on platelets from blood collected at 22˚C (blue symbols), 37˚C (pink), and exposed to 4˚C (green). Samples were either treated with vehicle (DMSO, filled symbols) or TRPM8 inhibitor PF 05105679 (2 μM, abbreviated as PF, open symbols). Samples were isolated from the blood of healthy volunteers (n = 3 to 5) who denied taking any platelet function-modifying medications for 14 days prior to the experiment. Analyses were performed using flow cytometry on LSR II. Platelets in PRP were distinguished by expression of CD61 using anti-CD61 PerCP-conjugated antibody. For each sample, 10,000 CD61 positive events were acquired. CD61 positive events were identified by gating approximately 1% of the appropriate isotype antibody in the positive gate. **B.** Percent of platelet microaggregates in PRP samples analyzed using imaging flow cytometry, n = 4. (**C** and **D**) representative images of two groups of platelet shape (**C**—spheroid cells, **D**—discoid cells). Samples were stained against integrin αIIbβ3 (PAC-1, green), p-Selectin (CD62P, yellow), and CD61 (red) fluorescent antibodies, and analyzed by imaging flow cytometry. The scale bar is 7 μm. (**E** and **F**). Percent spheroid (**E**) and discoid (**F**) shaped platelets in the PRP samples from blood collected at 22˚C (blue symbols), 37˚C (pink), and exposed to 4˚C (green), n = 4. Samples were analyzed using imaging flow cytometry. Platelets were assigned spheroid or discoid shapes as described in (Özpolat et al., 2023). Lines connecting data points indicate the same donor. Statistical analysis was performed using paired Student t-test, where asterisks indicated a p-value lower than 0.05 for *, 0.01 for **, and 0.005 for *** respectively, and "ns" indicates a p-value >0.05. Symbols above bars indicate paired comparison between treatment groups, and without bars indicate comparison to vehicle.

We hypothesized that cooling blood from 37˚C to 22˚C leads to a decrease in discoid events, and an increase in spheroid events, consistent with platelet activation. For samples collected at 37˚C, the fraction of spheroid-shaped platelets was 20.9 ± 2.8% (Mean ± SEM) and discoid cells were 78.4 ± 3.0% (Fig 3C and 3D), consistent with minimal PAC-1 binding (Fig 3A). Platelets that were kept at body temperatures had significantly larger spheroid shape fractions, than those at room temperature (Fig 3E, Mean ± SEM: 37.8 ± 2.6%, *p* = 0.004). This is in line with the observation that platelets are activated when cooled down from 37˚C to room temperature. Decreasing the temperature to 4˚C caused a further increase in the percentage of spheroid platelets (Fig 3E). Consequently, we observed a significant reduction in discoid platelets when they were exposed to 22˚C (Fig 3F, blue) and 4˚C (Fig 3F, green).

We hypothesized that room temperature-induced platelet shape change is TRPM8-dependent. For samples collected at 37˚C treated with PF 05105679, neither spheroid nor discoid populations were different from vehicle control (Fig 3E and 3F Mean ± SEM: 25.8 ± 4.4%, *p* = 0.2; and 73.4 ± 4.4%, *p* = 0.2 respectively). In contrast, pretreatment with PF 05105679 before exposure to 22˚C significantly reduced spheroid-shaped cell fraction and significantly increased discoid-shaped fraction compared to vehicle (Fig 3E and 3F 34.7 ± 2.7%, *p* = 0.0003; and 64.2 ± 2.8% p<<0.0001 respectively).

We also evaluated the effects of the short-term cooling of platelets to 4˚C. Blood collected at 37˚C and then incubated at 4˚C for 15min, evinced a significant increase in the spheroid fraction of platelets (Fig 3E 61.6 ± 1.9%, *p* = 0.003). This result is in alignment with previous reports describing a platelet spheroid shape change following cold exposure [3, 34]. However, 4˚C temperature effects were not significantly prevented by PF 05105679, either for spheroid (Fig 3E, 58.1 ± 2.9%, *p* = 0.1), or discoid fraction (Fig 3F, 40.6 ± 2.2%, *p* = 0.09). Together, these data confirm that a decrease in temperature activates platelets and that TRPM8 inhibition was sufficient to prevent platelet shape change after 22˚C exposure. However, the effects of exposure to 4˚C are not sufficiently prevented by TRPM8 inhibition, and likely involve other mechanisms.

## Platelets cannot be activated by TRPM8 agonists

Thus far, we have only activated TRPM8 by temperature and we next explored the effect of platelet TRPM8 activation by agonists. We briefly incubated washed platelets from 22˚C and 37˚C blood with different TRPM8 agonists and evaluated αIIbβ3 activation by PAC-1 binding using flow cytometry (Fig 4A). We did not observe a significant effect of either menthol, WS-12, or icilin on αIIbβ3 integrin activation at both temperatures tested (Fig 4A). Furthermore, pre-incubation with PF 05105679 did not affect the αIIbβ3 activation in the vehicle or TRPM8 agonist-treated platelets (Fig 4A).

Platelet activation can also be measured by α-granule secretion with a P-selectin (CD62P) antibody [35]. P-selectin positivity was not increased upon treatments with TRPM8 agonists compared to vehicle at either temperature (Fig 4B). In addition, pre-incubation with TRPM8 inhibitor PF 05105679 had no effect (Fig 4B). Furthermore, there was no platelet activation when we incubated platelets with TRPM8 agonists for 1 or 4 hours at either 22˚C or 4˚C (S5 and S6 Figs). Therefore, in platelets, activation of TRPM8 by chemical agonists induced neither αIIbβ3 activation nor degranulation of α-granules.

## Platelet aggregation is TRPM8-dependent in a subpopulation of donors

Next, we investigated whether TRPM8 activation affects platelet aggregation. We compared the amplitude of ADP-induced aggregation after the addition of TRPM8 agonists or the vehicle. At 37˚C TRPM8 agonists did not significantly enhance ADP-induced aggregation (Fig 5A,

**10 MINUTES INCUBATION**

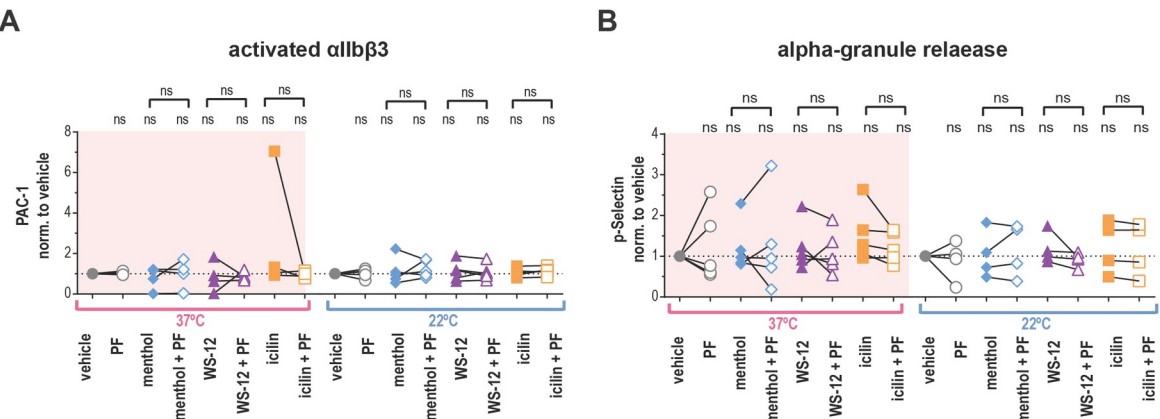

**Fig 4. TRPM8 agonists do not lead to integrin αIIbβ3 activation or P-selectin externalization in human washed platelets. A**) Samples were stained with PAC-1 (MFI normalized to the vehicle and (**B**) P-selectin (MFI normalized to the vehicle fluorescent antibodies, and evaluated via flow cytometry. First samples were pre-incubated with either vehicle DMSO or PF 05105679 (2 μM) for 5 minutes. Next, samples were treated with either vehicle (Ethanol), menthol (500 μM), WS-12 (2 μM) or icilin (100 μM) for 10 minutes at either 37˚C (pink background) or 22˚C temperature (white background). Values were normalized to those measured in platelets treated with vehicle. Lines connecting data points indicate the same donor. Statistical analysis was performed using paired Student t-test, where asterisks indicated a p-value lower than 0.05 for *, and "ns" indicates s p-value >0.05. Symbols above brackets indicate paired comparison between treatment groups, and without bars indicate comparison to vehicle.

pink background). In contrast, after 22˚C collection, treatment with menthol resulted in significantly higher aggregation compared to vehicle (Fig 5A, white background, $p = 0.001$). Interestingly, two out of six donors exhibited exceptionally enhanced aggregation when treated with WS-12 (representative traces in Fig 5B). On average, treatment with WS-12 resulted in a trend for increased ADP-induced aggregation to a degree that approached statistical significance ($p = 0.06$, see Fig 5A). Furthermore, when 22˚C platelets were pre-treated by the TRPM8 inhibitor PF 05105679 aggregation was inhibited in a subpopulation of donors (Fig 5A, white background, $p = 0.09$). In addition, PF 05105679 alone slightly decreased the ADP-induced platelet aggregation at 37˚C, but this effect also did not reach statistical significance (Fig 5A, $p = 0.06$). These data suggest a donor-dependent effect of TRPM8 activation on platelet aggregation.

## Platelet aggregation is reduced by TRPM8 inhibitor

We observed that TRPM8 inhibition reduces platelet activation, shape change, and aggregation. Therefore, we evaluated the effects of TRPM8 inhibition on aggregation initiated by different signaling pathways. We tested two specific TRPM8 inhibitors, PF 05105679 and AMTB [36], on platelets from room-temperature collections. We focused on two distinct pathways for the initiation of platelet aggregation–purinergic receptors (activated by ADP), and the immunoreceptor tyrosine-based activation motif (ITAM) receptor GPVI (activated by collagen and convulxin) [37, 38]. Neither PF 05105679 nor AMTB affected ADP-induced aggregation (Fig 5C). In contrast, both PF 05105679 and AMTB significantly inhibited collagen-induced aggregation (Fig 5D). In addition, treatment with PF 05105679, but not AMTB, significantly reduced convulxin-induced aggregation (Fig 5E). This indicates, that TRPM8 might be specifically involved in GPVI-mediated aggregation.

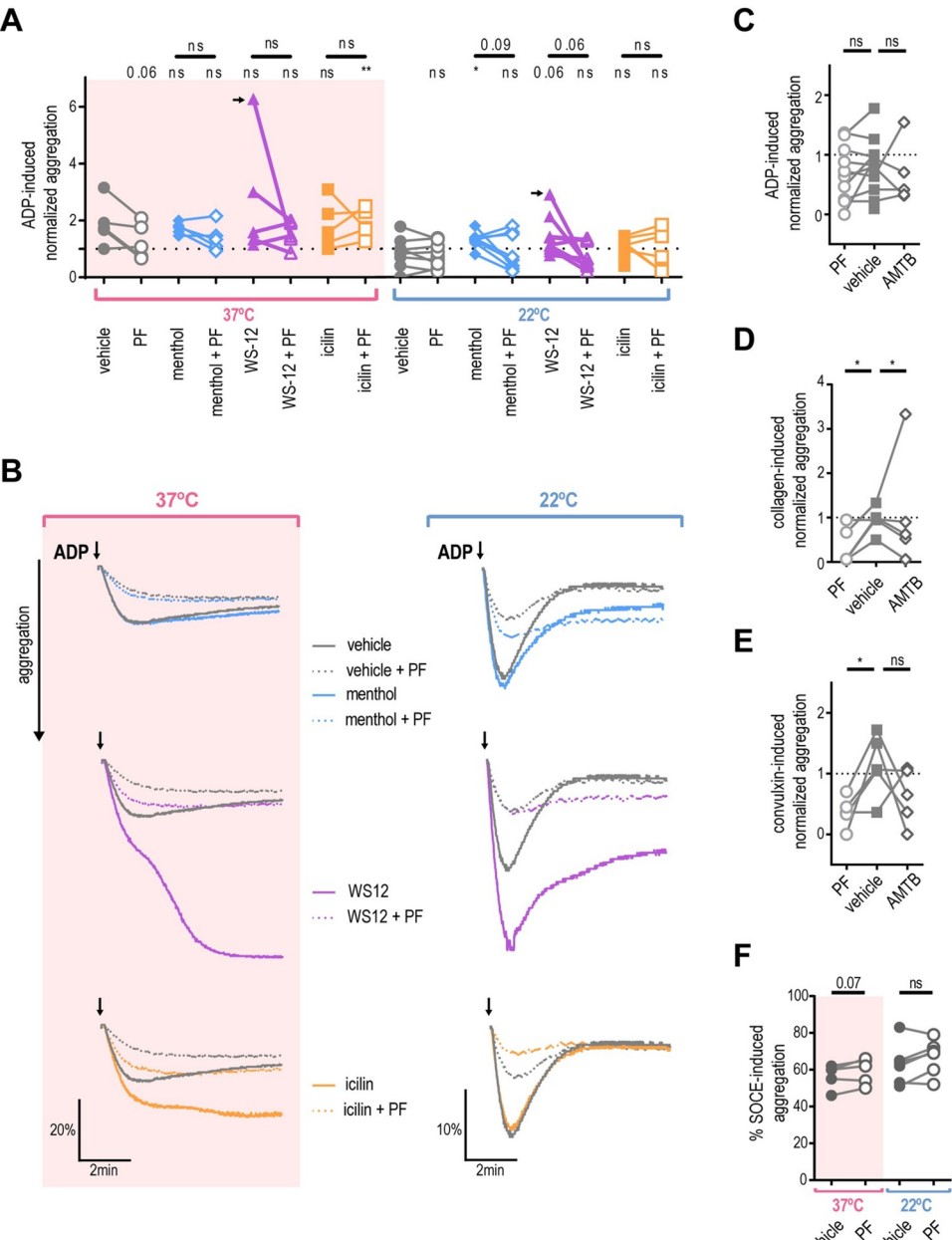

**Fig 5. Platelet aggregation is not TRPM8-dependent.** PRP isolated from blood collected by venipuncture at 37˚C (pink background) or 22˚C (white background) was subjected to light transmission aggregometry measured with stirring at 37˚C. **A.** ADP-induced aggregation normalized to ADP-only maximum (dotted line indicates 1). PRP from healthy donors was pre-treated with either vehicle DMSO (filled symbols) or PF 05105679 (2 μM, open symbols) for 5 minutes, then treated with either vehicle ETOH, menthol (500 μM, blue), WS-12 (2 μM, purple) or icilin (100 μM, orange) in presence of 1mM $Ca^{2+}$ for 5 more minutes. Subthreshold ADP concentration (previously identified for each donor (0.8 to 4 μM) as less than 60% maximal aggregation) was added and maximal aggregation was measured (for 37˚C n = 6, for 22˚C n = 10). Arrow indicates a hyper-responsive subject represented in B. **B.** Aggregation over time traces for the donor with especially strong WS-12 response. Subthreshold ADP was added at the time indicated by arrows. **C.** Effect of TRPM8 inhibitors on ADP-induced aggregation. PRP was pre-treated with either vehicle DMSO (filled squares), PF 05105679 (2 μM, open circles, n = 10) or AMTB (10 μM, open diamonds, n = 5). **D.** Effect of TRPM8 inhibitors on collagen-induced aggregation (n = 5). Subthreshold collagen concentration (previously identified for each donor (0.25 to 1 μg/ml) as less than 60% maximal aggregation). Values were normalized to the maximum aggregation achieved by collagen alone (dotted line). **E.** Effect of TRPM8 inhibitors on the convulxin-induced aggregation (n = 5). Subthreshold convulxin concentration (previously identified for each donor (1 to 10 ng/ml) as less than 60% maximal aggregation). Values were normalized to the maximum aggregation achieved by

convulxin alone (dotted line). **F.** Effect of TRPM8 inhibitors on the SOCE-induced aggregation (n = 5). PRP was pre-treated with thapsigargin (5 μM) for 10 minutes prior to the addition of 1 mM Ca$^{2+}$ (no other aggregation agonists were used). Raw percent aggregation values are reported. Lines connecting data points indicate the same donor. Statistical analysis was performed using paired Student t-test, where asterisks indicated a p-value lower than 0.05 for *, and "ns" indicates s p-value >0.05. Symbols above bars indicate paired comparison between treatment groups, and without bars indicate comparison to vehicle.

We next investigated whether TRPM8 contributes to store-operated calcium entry (SOCE) during aggregation. To isolate the contribution of SOCE to aggregation, we first emptied the calcium stores with the Sarco-Endoplasmic Reticulum Calcium ATPase (SERCA) inhibitor thapsigargin in the absence of external Calcium [39]. Platelets were also treated with either PF 05105679 or vehicle and SOCE–induced aggregation was initiated by the addition of 2 mM Ca$^{2+}$. There was no statistically significant effect of PF 05105679 treatment on SOCE-induced aggregation (Fig 5F), although a trend to an increase at 37˚C was evident (Fig 5F, $p = 0.07$). This indicates that TRPM8 inhibition does not affect the intrinsic SOCE response of platelets.

## Platelets do not exhibit TRPM8-dependent calcium influx

To evaluate the role of TRPM8 for cytosolic calcium influx, we recorded fluorescence from Fura 2-AM loaded platelets using a microplate spectrophotometer. Fig 6A shows the average calcium levels over time from several donors in response to the application of menthol. The addition of the vehicle induced a modest decrease in baseline calcium of 17.1 ± 6.3 nM for 37˚C and 7.2 ± 4.7 nM for 22˚C (Fig 6A and 6B), which was probably due to the disturbance of the platelet suspension by the mixing. In comparison, the addition of menthol resulted in a greater rise in calcium levels: 33.0 ± 15.5 nM for 37˚C and 19.7 ± 7.0 nM for 22˚C (Fig 6A and 6B). At 22˚C the addition of menthol tended to increase calcium levels compared to the vehicle (approaching significance $p = 0.07$, Fig 6B), however, this effect was absent at 37˚C ($p = 0.17$). The addition of WS-12 or icilin at 22˚C or 37˚C did not result in a significant change in calcium compared to the vehicle (Fig 6B).

Next, we investigated whether inhibition of TRPM8 resulted in changes in the calcium response to agonists. Surprisingly, pre-treatment with the TRPM8 inhibitor PF 05105679 reduced the calcium response as compared to vehicle addition at 37˚C (Fig 6B, $p = 0.02$). Furthermore, the inhibitory effect of PF 05105679 was also statistically significant in menthol-treated platelets at 37˚C (Fig 6B, $p = 0.04$). For ADP, pre-treatment with PF 05105679 led to a significant decrease in the calcium response at 37˚C and 22˚C (Fig 6C, $p = 0.03$ and 0.05 respectively). In contrast to our aggregometry findings, we observed that calcium response to high concentration convulxin was not inhibited by PF 05105679 as compared to vehicle (Fig 6D, $p = 0.2$ for 37˚C and 0.7 for 22˚C). These findings suggest that there might be some effect of TRPM8 inhibition on platelet calcium homeostasis.

Finally, we investigated if TRPM8 inhibition affected the store-operated calcium entry in platelets by directly measuring calcium influx after SOCE. Due to the large amplitude of SOCE responses, we loaded the washed platelets with Fluo 4-AM, the dye which has a lower affinity to calcium compared to Fura 2-AM. To empty the calcium stores before SOCE induction, platelets were re-suspended without extracellular calcium and treated with thapsigargin. Simultaneously, platelets were either treated with vehicle or PF 05105679. SOCE was induced by the addition of 2 mM calcium to the platelet suspension. Maximal calcium influx was unaffected by PF 05105679 at both temperatures (Fig 6E). Thus, in line with the aggregometry findings, we show that TRPM8 inhibition does not affect the intrinsic SOCE properties of platelets.

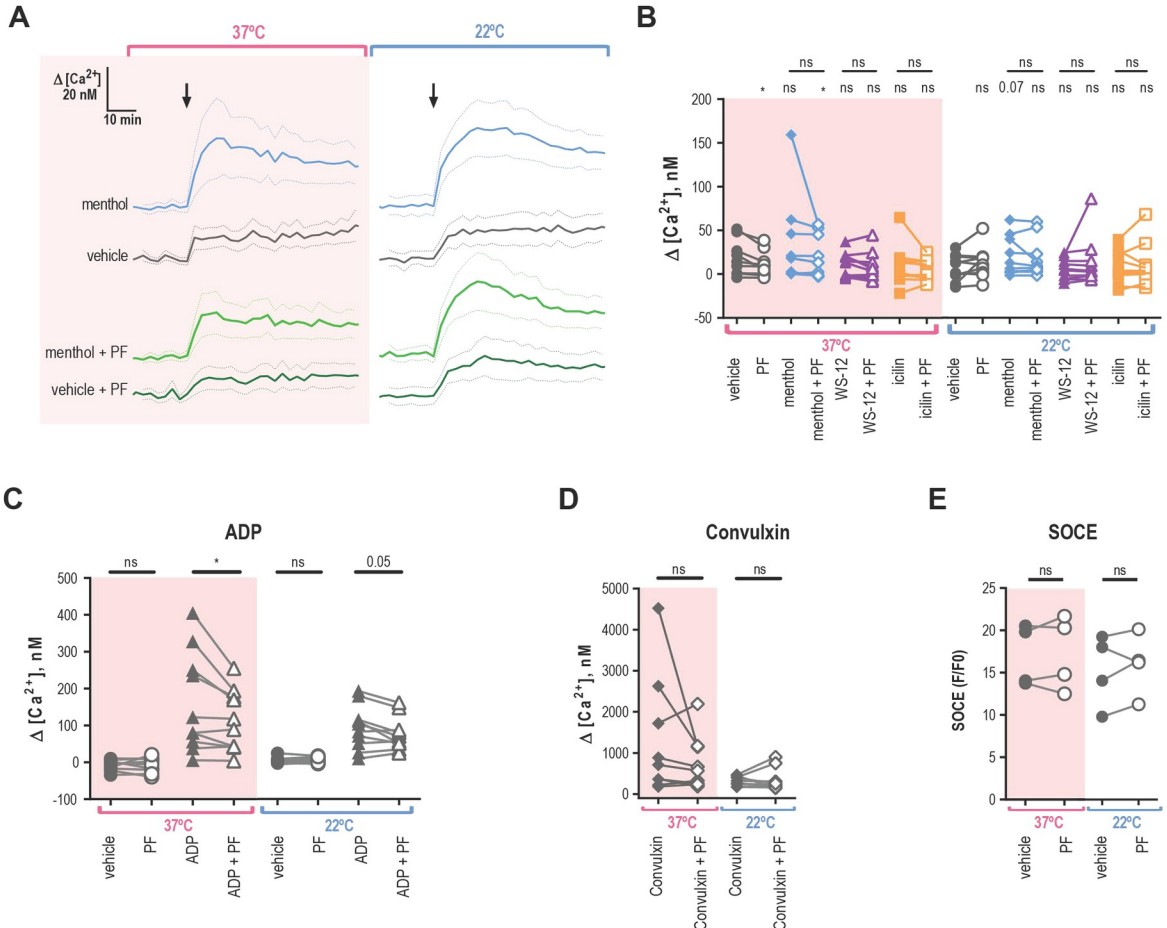

**Fig 6. Platelets do not elicit TRPM8-dependent calcium influx.** Change in intracellular calcium of washed platelets at different temperatures (pink background for 37°C and white background for 22°C), loaded with Fura 2-AM (A-D) or Fluo 4-AM (**E**), in the presence of 2mM extracellular $Ca^{2+}$. **A.** Average change in calcium concentration over time in washed platelets treated with menthol (500 μM) or vehicle ETOH pre-treated by the vehicle DMSO or PF 05105679 (2 μM). **B.** Maximal change in calcium concentration after addition of vehicle ETOH (gray), menthol (500 μM, blue), WS-12 (2 μM, purple), or icilin (100 μM, orange) when pre-treated by the vehicle DMSO (filled symbols) or PF 05105679 (2 μM, open symbols). **C.** Maximal change in calcium concentration after addition of vehicle (circles) or ADP (20 μM triangles), while pre-treated by the vehicle DMSO (filled symbols) or PF 05105679 (2 μM, open symbols). **D.** Maximal change in calcium concentration after the addition of convulxin (100ng/ml), while pre-treated by the vehicle DMSO (filled symbols) or PF 05105679 (2 μM, open symbols). Star symbols above each dataset indicate a comparison to the corresponding vehicle shown in C. **E.** SOCE-induced maximal change in baseline-normalized Fluo 4-AM fluorescence after the addition of 2mM extracellular $Ca^{2+}$, while pre-treated by the vehicle DMSO (filled symbols) or PF 05105679 (2 μM, open symbols, n = 4). Lines connecting data points indicate the same donor. Statistical analysis was performed using paired Student t-test, where asterisks indicated a p-value lower than 0.05 for *, 0.01 for **, and 0.005 for *** respectively, and "ns" indicates a p-value >0.05. Symbols above bars indicate paired comparison between treatment groups, and without bars indicate comparison to vehicle.

## Acute cold exposure leads to immediate ER-derived calcium increase independent of TRPM8

We investigated the effect of fast chilling on platelet intracellular calcium levels using the temperature-controlling capabilities of a real-time PCR detection system. We selected the calcium-sensitive fluorescent dye Calcium Green™-1, AM for its lack of temperature-dependent changes in brightness [40]. The cells were accustomed to the 37°C temperature and then cooled rapidly to 10°C while recording the calcium signal. TRPM8 expressing HEK293T/17 cells exhibited temperature responses typical of TRPM8, which were inhibited by PF 05105679 or the absence of extracellular Calcium (S7A–S7C Fig). Next, we evaluated calcium response

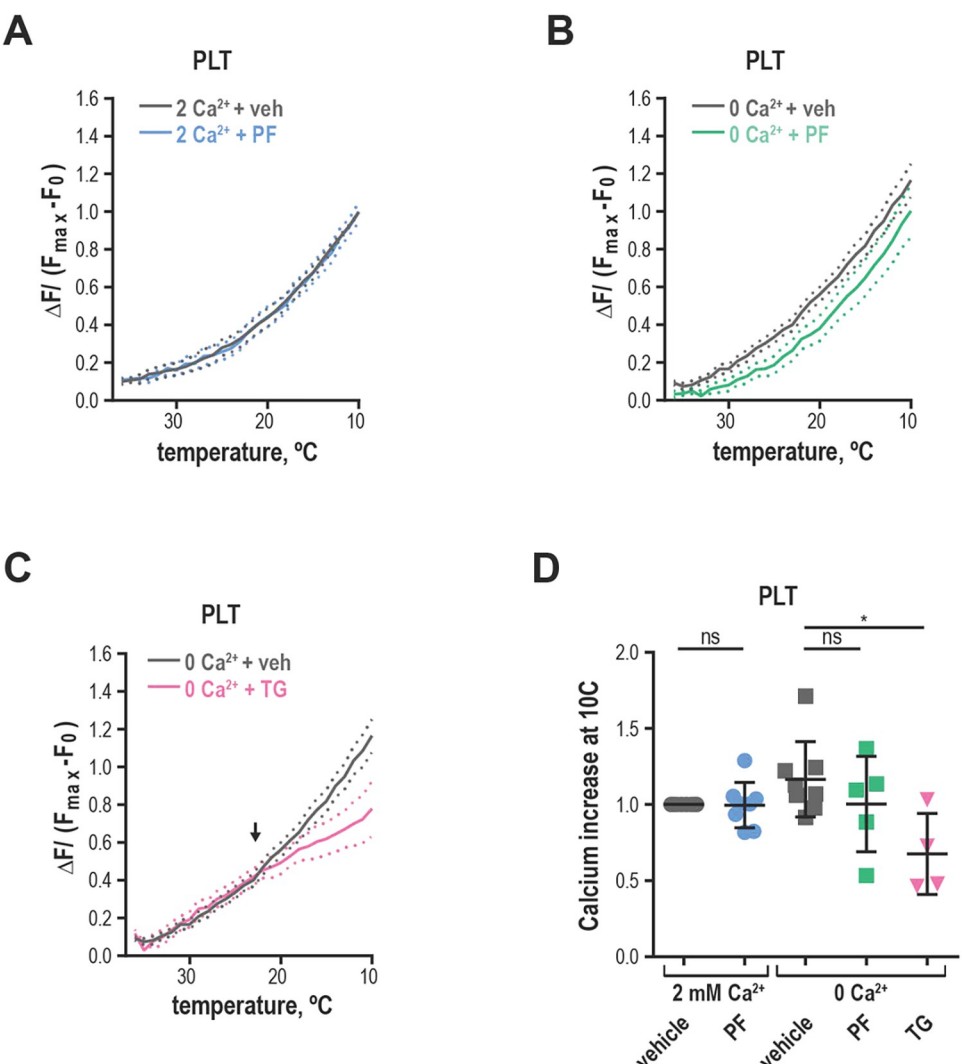

**Fig 7. Chilling platelets leads to a rapid calcium increase independent of TRPM8. (A-B)** Change in Calcium Green™-1 fluorescence levels baseline subtracted and normalized to maximum obtained after addition of calcium ionophore 7 μM A23187. **A.** Washed platelets in 2 mM $Ca^{2+}$ Tyrode's buffer with either vehicle DMSO (gray) or 2 μM PF 05105679 (blue). **B.** Washed platelets in 0 mM $Ca^{2+}$ and 100 μM EGTA -containing Tyrode's buffer with vehicle DMSO (gray) or 2 μM PF 05105679 (green). **C.** Washed platelets in 0 mM $Ca^{2+}$ and 100 μM EGTA -containing Tyrode's buffer with vehicle DMSO (gray) or 5 μM thapsigargin (pink). Arrow indicates an apparent threshold for platelet activation at ~ 23˚C. **D.** Quantification of maximal calcium increases at 10˚C in washed platelets from 4 to 8 donors. Error bars indicate Mean ± SEM. Statistical analysis was performed using paired Student t-test, where asterisks indicated a p-value lower than 0.05 for *, and "ns" indicates a p-value >0.05. Symbols above bars indicate paired comparison between treatment groups.

to chilling in washed human platelets. We observed a curved increase in fluorescence when extracellular $Ca^{2+}$ was present (Fig 7A). The addition of TRPM8 inhibitor PF 05105679 did not affect the amplitude or shape of the temperature response in platelets in the presence of $Ca^{2+}$ (Fig 7A). Furthermore, when $Ca^{2+}$ was omitted, PF 05105679 did not affect temperature response curves (Fig 7B). These data suggest that acute temperature-induced calcium influx in platelets is not mediated by TRPM8.

Because we showed that platelets exhibit an increase in intracellular calcium in response to chilling even in the absence of extracellular calcium, we surmised that internal calcium stores were responsible for this phenomenon. To test this hypothesis, we emptied the stores by

treating the platelets with thapsigargin in the absence of extracellular calcium before temperature stimulation. Thapsigargin-treated temperature response curve no longer resembled that of control platelets (Fig 7C) and was similar to a linear response of untransfected HEK293T/17 cells in the presence of $Ca^{2+}$ (S7B Fig). The responses from untransfected HEK293T/17 cells and platelets treated with thapsigargin share the same fit curve (S7D Fig, the extra sum of squares F-test using Prism: $p = 0.98$, f = 0.02). In addition, in thapsigargin-treated platelets the maximal calcium response at 10˚C was significantly lower than that of a vehicle (Fig 7D, $p = 0.04$). Furthermore, platelets exhibited a clear threshold for activation by the temperature below ~23˚C (Fig 7D, arrow). We conclude that acute response to chilling in platelets is dependent on the calcium released from the endoplasmic reticulum (also referred to as a dense tubular system, DTS).

## Discussion

In this study, we investigated the ion channel TRPM8 in platelets and megakaryocytes. TRPM8 was found on monocytes and lymphocytes in the past and sequencing data hinted at its presence in platelets. As expected, we found evidence for it in the megakaryocytic cell line MEG-01 and platelets. MEG-01s, like megakaryocytes, shed platelet-like particles and are in many other ways akin to primary megakaryocytes, although they only represent one maturation stage [41, 42]. Even though we observed some TRPM8-dependent calcium influx in megakaryocytes, we did not extensively investigate the role of TRPM8 in megakaryocyte function which should be a subject for future studies.

In platelets, we focused on TRPM8 expression at the protein level. We showed via western blot, that some donor platelet lysates have a very faint band corresponding to the full-length TRPM8 protein. Surprisingly, in most donors, we observed a variety of more abundant smaller TRPM8 proteins, including less characterized shorter isoforms of TRPM8 [43]. In particular, the ~45kDa isoform was previously reported to be functional in the endoplasmic reticulum of prostate primary epithelium cells and keratinocytes [44, 45]. These shorter-length TRPM8 isoforms may not be delivered to the plasma membrane and, therefore, be undetectable by the immunostaining of non-permeabilized platelets. In addition, very short isoforms (8 and 16kDa) were shown to inhibit and stabilize the closed conformation of full-length TRPM8 [46]. One possible explanation is that the remaining full-length TRPM8 ion channels on the plasma membrane are completely inhibited or desensitized in resting platelets, but this needs further testing [47]. This is further supported by our counterintuitive finding that the majority of TRPM8-positive platelets have a discoid shape (resting). Notably, TRPM8 has recently been shown to exhibit a non-channel function, by affecting endothelial cell adhesion and motility by directly interacting with signaling molecules [48].

Platelets presented a punctate expression pattern of TRPM8, which is in alignment with findings from other cell types [29]. An alternative explanation for this finding is that the TRPM8 protein source is from adherent microparticles e.g., of leukocyte origin. However, although incubation of platelets with CD45+ leukocyte microparticles increased TRPM8 expression, the staining did not necessarily overlap with TRPM8 on platelets. We did not exclude endothelial cells as the source for TRPM8-positive microparticles. In contrast, MEG-01 cells were obtained from cell culture and were therefore devoid of any other cellular contamination. Overall, microparticles from different other cell types likely contribute to TRPM8 expression but are unlikely the sole source of TRPM8 in platelets.

Platelets become activated at room temperature [23]. After having shown TRPM8 on megakaryocytes, we hypothesized that it is a cold sensor on platelets and its inhibition circumvents the platelet activation response to below body temperature. As expected, TRPM8-specific

inhibitor PF 05105679 [24, 25] leads to a significant decrease in room temperature-induced platelet shape change. Surprisingly, in another assay, the decrease in integrin activation only approached statistical significance. Our recently developed assay for platelet shape change detection by imaging flow cytometry [34] may be especially sensitive in detecting the early stages of shape change and platelet activation. In addition, TRPM8 may be more relevant for shape change than for integrin activation and other cold-dependent signals may contribute to αIIbβ3 activation. Indeed, when platelets are stimulated by agonists, shape change facilitates adhesion [49] and precedes αIIbβ3 integrin activation and aggregation [50]. At the site of superficial vascular injury, blood could be exposed to colder temperatures. Therefore, an immediate, cold-induced shape change could facilitate platelet adhesion to subendothelial matrix proteins such as collagen and thereby play a physiological role in hemostasis. However, we cannot exclude that the PF 05105679 inhibitor has off-target effects on other signaling pathway components upstream of SOCE.

We expected TRPM8 to be excitable by chemical agonists as has been reported in other cell types. We tested three different chemical agonists of TRPM8, namely menthol, WS-12, and icilin in multiple assays. Surprisingly, they consistently failed to cause platelet activation. In only one assay, menthol, but not other TRPM8 agonists, lead to a significant increase in aggregation. Menthol is the least specific of the agonists we used, and it was shown to have TRPM8-independent responses in other cell types [51]. We used different ambient temperature protocols because TRPM8 activation is enhanced by temperature, which may explain why at 22°C (but not at 37°C) the calcium increase to menthol was approaching significance. In addition, the low intensity of TRPM8-positive platelets observed via flow cytometry combined with faint signals on western blot, suggests that there is a small number of copies of TRPM8 per cell. The low prevalence of TRPM8-positive platelets may lead to a small response amplitude of the population measurements, bringing it close to the detection limit of the assays used in this study. Overall, our data indicate that platelet TRPM8 appears unresponsive to chemical agonists, at least with acute stimulation.

One of the hallmarks of platelet activation by temperature is the increase in cytosolic calcium. The pathophysiologic mechanism for this has been elusive, but multiple non-exclusive mechanisms have been proposed for this phenomenon, such as an imbalance in the activity of the $Ca^{2+}$-ATPases, the phase transition of lipids, the changes in PLC activity, or the calcium release from the ER. Consistent with the previous report by [15], we observed that platelets exhibit a rapid increase in cytosolic calcium in response to lowered temperatures. Given its role as a calcium-permeable ion channel, we expected TRPM8 to play a critical role in this phenomenon. However, TRPM8 was completely dispensable for the temperature-induced calcium increase. In addition, for the first time to our knowledge, we identified that the rapid increase in intracellular calcium is directly dependent on the ER calcium stores [15, 16, 40, 52, 53]. Further investigation is needed to identify the molecular entity responsible for the temperature-induced calcium release from the ER.

Several prospects arise from our findings. First, further studies are needed to identify the functional role of TRPM8 expression in megakaryocytes. Equally intriguing is the identification of the molecular mechanism for cold sensation at the endoplasmic reticulum level. This would allow to design a pharmaceutical intervention to interfere with this acute response, e.g., in patients with therapeutic hypothermia or to inhibit the cold response of stored platelets. Our study also highlights the need to understand the large variability in TRPM8 expression on a healthy volunteer population basis.

In conclusion, TRPM8 contributes to some features of the acute cold response in platelets but is dispensable for acute cold-induced calcium increase. Instead, we identified a critical role of ER-dependent calcium in response to cold.

## Materials and methods

### Cell culture and transfection

MEG-01 cells were obtained from ATCC (#CRL-2021). Meg-01 cells were maintained at 37˚C and 5% $CO_2$ in 25cm$^2$ cell culture flasks with 10ml of cell culture media containing DMEM (#11995–065, Sigma-Aldrich, St. Louis, MO) and 10% FBS (#30–2020, ATCC, Manassas, VA) without antibiotics. The cells were propagated by scraping off the flask bottom using a cell scraper, transferring to a conical tube, and centrifuging at ~300g (1100rpm) for 7min. The cell pellet was re-suspended in a small volume of fresh media and diluted at 1:2 or 1:3 into a fresh pre-warmed culture media. The cell propagation was performed twice a week. For not more than ~20 propagations.

HEK293T/17 cells originally from ATCC Cat # CRL-11268 were obtained from Sharona Gordon Lab (University of Washington, Seattle). Cells were maintained in culture at 37˚C and 5% $CO_2$ in 10cm plates with 10ml of media containing DMEM (#11995–065, Sigma-Aldrich, St. Louis, MO) and 10% FBS (#30–2020, ATCC) and Pen/Strep (Cat# 30-002CL, Corning). The cells were propagated by treatment with PBS+ 2mM EGTA for 5 min, trituration, and dilution at 1:20 twice a week.

HEK293T/17 cells were transfected according to the manufacturer's protocol using Lipofectamine 2000 (#11668019, ThermoFisher, Waltham, MA) with either pEGFP_TRPM8 (#64879, Addgene, Watertown, MA) or pEGFP-C1 (Clonthech Laboratories, Mountain View, CA). Cells were visually inspected for GFP expression using a fluorescent microscope and used for experiments after 24–48 hours post-transfection. The experimental buffer used for HEK293T/17 cells was Hepes buffered Ringers (HBR): made of double deionized water and in mM: 140 NaCl, 4 KCl, 1 MgCl2, 1.8 CaCl2, 10 HEPES (free acid) and 5 glucose).

### Calcium imaging

MEG-01 cells were seeded in a small droplet with a high density of cells onto poly-L-lysine (#P1524, Sigma-Aldrich, St. Louis, MO) treated 25mm diameter round glass coverslips placed in 6 well plates for ~1hr on the day of the experiment. After the time for attachment elapsed, wells were flooded with culture media containing calcium-sensitive dye 3uM Fluo4-AM (#F14201, ThermoFisher, Waltham, MA) and 0.2% Pluronic F-127 (# P3000MP, Thermo-Fisher, Waltham, MA) for 1hr in the dark at 37˚C in 5%$CO_2$ atmosphere. After loading, coverslips were washed with Modified Tyrode's Buffer (MTB, 137 mM NaCl, 0.3 mM $Na_2HPO_4$, 2mM KCl, 12mM $NaHCO_3$, 2mM $CaCl_2$, 5mM HEPES, 5mM glucose, pH 7.3) for additional 1 hour to allow de-esterification of Fluo4-AM. Coverslips were placed in the bottomless home-made chamber with a Teflon round gasket with a round cut-out. The resulting volume of the chamber was ~ 1 ml and was slowly exchanged at ~0.3ml/min rate using a gravity-driven homemade system. MEG-01 cells were allowed to adjust to room temperature and flow of buffer for ~15min prior to recording. Individual intracellular calcium levels of MEG-01 cells were recorded using an inverted XI81F-3 microscope (Olympus, equipped with a motorized stage (MS-2000, (Applied Scientific Instrumentation, Eugene, OR), fluorescent Xenon lamp (LB-XL, Sutter Instruments, Novato, CA), FITC excitation emission filter set, automated shutter, 10x objective and a CCD camera (C4742-80-12AG, Hamamatsu, Japan). Images were acquired using SlideBook software (Intelligent Imaging Innovations, Inc., Denver, CO) every 5 seconds with 500 ms exposure. Each recording was no longer than 40 minutes. The data was analyzed using FIJI software. Region of interest (ROI) was drawn around each cell and average pixel intensity over time was measured. Background signal from a similarly sized ROI without cells was subtracted from each cell signal. Individual cell intensity was normalized according

to the formula: $F/(F_{max}-F_0)$, where F is the average fluorescence of an ROI at any given time, $F_{max}$ is a maximum intensity measured during the calcium-ionophore 7 μM A23187 (#C7522, Millipore Sigma, Burlington, MA), application at the end of every recording, and $F_0$ is the baseline fluorescence before the addition of agonists. An 0.025 upward deflection in normalized Fluo4 fluorescence was considered a positive response. Only the ADP–responsive cells were included in the analysis (Fig 1F, 94% positive cells).

## Isolation of mRNA and PCR amplification of TRPM8 sequences

Messenger RNA was isolated by harvesting $\sim10^6$ cells in Trizol reagent and chloroform (at a 5:1 ratio). Samples were then centrifuged at 12000g for 15 min at 4˚C. RNA from the supernatant was precipitated by the addition of isopropanol and centrifuged again. RNA pellet was washed once with 75% Ethanol, mixed, and centrifuged at 7500g for 5min at 4˚C. RNA pellet was dried and then re-suspended in water.

RACE-ready (Rapid Amplification of cDNA 5' Ends) cDNA was generated using SMARTer PCR cDNA Synthesis Kit (#634925, Clontech Laboratories, Inc.) according to the manufacturer's protocol.

TRPM8 sequences were amplified by PCR using the following primers (ordered from Millipore Sigma, Burlington, MA):

1410F forward primer 5'–CCCAAGTTTGTCCGCCTCTTTCTGGAG–3'
1788R reverse primer 5'–CAGAAGCTTGCTGGCCCCCAAGGCTGC–3'
2017R reverse primer 5'–CAGGCTGAGCGATGAAATGCTGATCTG–3'

PCR amplification was carried out using a Q5® High-Fidelity DNA Polymerase kit (#M0491L, New England Biolabs, Ipswich, MA) using the following protocol: 40 cycles of 30-sec denaturation at 98C, then 30-sec annealing at 63C and 1min amplification at 72C.

PCR products were subjected to agarose gel electrophoresis along with 1Kb Plus Ladder™ (#10787–018, Invitrogen, Waltham, MA) standard. The bands of expected sizes were cut out of the agarose gel and extracted using QIAquick Gel Extraction Kit (#28706, Qiagen, Hilden, Germany) according to the manufacturer's protocol.

After gel extraction, PCR products were cloned into a TOPO™ TA Cloning™ Kit for Subcloning, with Top10 E.coli competent cells (#K450001SC, ThermoFisher, Waltham, MA) according to the manufacturer's protocol. Competent cells were transformed with the resulting cloning vector and grown on the plates containing LB with Kanamycin (50ng/ml). The positive colonies were cultured and the DNA was isolated using the QIAGEN MiniPrep Kit (#27106, Qiagen). The resulting DNA was submitted for sequencing with Genewiz (South Plainfield, NJ).

## Ethics statement

The Western Institutional Review Board -Copernicus Group (WCG-IRB) approved the research, and all human participants gave written informed consent. They were asked to include date, time, and signature. Consent was witnessed and documented by a research coordinator, who also provided, the date, time, and signature. Humans were recruited to the Bloodworks Northwest Research Institute (Seattle, WA, USA) to provide blood samples from 1. September 2020 to 30. June 2023. The study was conducted following the Declaration of Helsinki.

## Blood collection

Blood was collected from healthy human volunteers denying any platelet function-modifying pathology or medications for at least 14 days prior to collection, according to the approved

protocol. Using a 21G or larger needle, the first 3 ml of blood was discarded. Then blood was collected into a syringe filled with ACD-A (1:6) with less than or equal to 20 mmHg tourniquet pressure. ACD-A in the syringe was pre-warmed to the 22˚C or 37˚C temperatures as indicated for each experiment. If the experiment indicated 37˚C collection–every step was performed at 37˚C, with very minimal exposure to room temperature.

## Platelets washing

After collection, blood was transferred into a conical tube and let to rest for at least 20 min before gentle centrifugation at 190g for 15 min with slow deceleration. Platelet-rich plasma (RRP) with a buffy coat (containing white cells) was transferred into a new tube and centrifuged gently for 5min for better PRP separation. The supernatant containing RPR was transferred to a new tube and incubated with 0.5uM $PGI_2$ (#P6188, Millipore) for 15 min. Extra ACD-A at 1:10 was added to PRP before the fast centrifugation step at 900g for 15min with slow deceleration. Platelets were re-suspended in Modified Tyrode's buffer without calcium (MTB-1, 137 mM NaCl, 0.3 mM $Na_2HPO_4$, 2mM KCl, 12mM $NaHCO_3$, 5mM HEPES, 5mM glucose, 0.35% HSA (w/v), 10 U/mL heparin, and 0.5 μM $PGI_2$, 0.2 U/mL apyrase, pH 7.3) [54], density was adjusted to 300K platelets/μL and incubated for 15 min. Next, ACD-A at 1:10 was added to washed platelets immediately prior to a second fast spin at 900g for 5 min. Finally, the pellet was re-suspended in Modified Tyrode's buffer (MTB-2, 137 mM NaCl, 0.3 mM $Na_2HPO_4$, 2 mM KCl, 12 mM $NaHCO_3$, 5 mM HEPES, 5 mM glucose, 0.1% HSA (NDC 68516-5216-1, GRIFOLS), 0.02 U/mL apyrase (#A6535, Sigma), pH 7.3). Platelet density was adjusted to 300K platelets/μL and $CaCl_2$ was added depending on the assay (see each experiment description in the Results section).

## CD45-positive cells depletion of PRP for platelet purification

PRP from human subjects was subjected to white blood cell (WBC) and red blood cell (RBC) depletion by using CD45 MicroBeads (for WBCs, #130-045-801, Miltenyi Biotec., Gaithersburg, MD) and CD235a (Glycophorin A) MicroBeads, (for RBCs, #130-050-501, Miltenyi Biotec.). PRP was incubated with beads for 20 minutes at RT, and then subjected to magnetic selection/depletion through autoMACS® Pro Separator (Miltenyi Biotec.).

## Western blot

Donor blood was collected at room temperature in a syringe containing ACD-A. After a resting period of 20 min, the samples were spun down at 200g for 15 min. PRP without a buffy coat was collected and incubated with PGI for 15min. Next, samples were spun at 900g for 15 min and the pellet was re-suspended in MTB-1. The washed platelets pellet was re-suspended in a 2x RIPA buffer (1% Nonidet P-40, 0.5% sodium deoxycholate, 0.1% SDS, 50 mM Tris base, 150 mm NaCl, EDTA-free Protease Inhibitor Cocktail (#11836170001, Roche) and pH 8.0) at $2 \times 10^6$ per $mm^3$ density and incubated with mixing for 30 minutes at 4˚C. Samples were centrifuged at 4˚C for 30 min at 14,000 ×g, the supernatant was transferred to another tube, and added SDS sample buffer with 5% BME. Samples were boiled for 10 min at 95C and subjected to NuPage 3–8% Tris-Acetate gel (#EA0375PK2, Invitrogen). Separated proteins were transferred to cellulose membrane using a semi-dry transfer method and blocked in TBS-T with 5% dry milk containing 2% normal goat serum for 1hr. Where specified, prior to staining, the primary anti-TRPM8 antibody was incubated with the corresponding blocking peptide (Cat # BLP-CC049, Alomone Labs, Israel) at a 1:1 ratio at room temperature for 1hr. Next, membranes were incubated with primary antibodies: anti-TRPM8 antibody (extracellular epitope, #ACC-049, Alomone Labs, Israel) at 1:400 dilution overnight at 4˚C. Next,

membranes were washed with TBS-T 5x times and incubated with secondary antibodies: Goat anti-Rabbit IgG (H+L) Secondary Antibody HRP antibody at 1:1000 (#31460, ThermoFisher) or Anti-Mouse IgG (#NA931, Amersham/ GE Healthcare Life Sciences, United Kingdom) at 1:30,000 for 1hr at RT. Proteins were visualized by subjecting membranes to SuperSignal™ West Femto Maximum Sensitivity Substrate (#34095, ThermoFisher) in the dark imager chamber equipped with a camera. Images were processed using FIJI software. For a line scan quantification, a line was drawn in each lane of the gels, and pixel intensity over distance was measured using the ROI Manager tool of FIJI. Each lane was normalized to the average pixel intensity inside an ROI drawn at the high molecular weight. Finally, measurements from line scans of the blots incubated with the blocking peptide were subtracted from the corresponding lanes of the blot stained with only primary and secondary antibodies.

## Conventional flow cytometry

To determine the levels of platelet activation at different temperatures, whole blood was collected by venipuncture using a vacutainer containing sodium citrate at 37˚C or 22˚C. PRP was isolated as described above and incubated for 30 minutes at the assigned temperature. PRP was adjusted to a platelet density of 300K platelets/µl with PPP, which was isolated by centrifugation at 2000g for 5 minutes. PRP was diluted with PBS (1/10 dilution) and stained with PAC-1 (FITC, BD #340507), CD61 (PerCP, BD, #340506) and P-selectin (PE, BD #555749) fluorescent antibodies or the corresponding isotypes Mouse IgM (FITC, BD #340755) and Mouse IgG1 (PE, BD #349044). The platelets were incubated with the antibodies for 15 minutes at a designated temperature and the samples were fixed with 500 µL paraformaldehyde (PFA, final concentration 1%).

To determine the levels of washed platelet activation, samples were first prepared as described above. Next, washed platelets were incubated in the presence of 1 mM $CaCl_2$ with the CD61 (PerCP), PAC-1 (FITC), and P-selectin (PE) and their corresponding Isotype controls in another tube for 15 min in the dark, while also incubating with TRPM8-specific agonists and/or the inhibitor. The reaction was stopped and the samples were fixed with the addition of a PFA (final concentration 1%).

*TRPM8 staining*: PRP diluted in PBS was incubated with CD41a (anti-human, APC, Invitrogen) and TRPM8 antibody (1:250 dilution, #ACC-049, Alomone) for 30 minutes at room temperature, and followed by the incubation with the secondary antibody (FITC- goat anti-rabbit IgG, 1:250 dilution, #L43001 Invitrogen) for 30 minutes at room temperature. The samples were fixed with PFA (final concentration 1%).

Samples were run on the LSR II flow cytometer (BD Biosciences). For each sample, a total of 100,000 events were acquired. Data were analyzed by FlowJo V10 software. Platelets were distinguished from other blood cells by using forward (FSC) and side scatter (SSC) and CD61 positive events. The gating strategy for PAC-1 and P-selectin was selected based on the binding of the appropriate isotype antibodies.

## Imaging flow cytometry

For TRPM8 positivity we used the same staining protocol as for conventional flow cytometry. We employed a gating strategy that excluded the TRPM8 (FITC) intensity of the unstained (Fig 2A) and secondary-only control samples (Fig 2B), to identify the TRPM8-positive population (Fig 2C). The samples were run on Amnis Imagestream Mk II (Luminex Corp, Austin, TX) image flow cytometer as described before [55, 56]. Briefly, the samples were acquired using the 60X magnitude objective and the following channels: the bright field (Ch01), the SSC (Ch06), FITC (Ch02), and APC (Ch11). For each sample, 20,000 events were collected using

the INSPIRE software. We used IDEAS 6.2 software (Luminexcorp, Austin, TX) for data analysis. Focused platelets identified by Gradient RMS channel M01 ($> 60\%$). Platelets were defined by using the CD61 (PerCP) or CD41 (APC) signal.

To evaluate the single platelet morphology and platelet micro-aggregates PRP was stained as described above. We used the 'Area' and the 'Aspect Ratio' features to analyze focused platelets. To distinguish micro-aggregates and single platelets, we used the Area (Area-Object M01, Ch01, Tight) feature (micro-aggregates $> 23$ μm$^2$, single platelets $\leq 23$ μm$^2$). Further, we define circular and fusiform platelets using the 'Aspect Ratio Intensity' (M01_Ch01) under the single platelet population (Circular platelets $> 0.8$ and fusiform platelets $\leq 0.8$) [34].

## Immunostaining

MEG-01 cells were attached to poly-L-lysine coated glass coverslips for 30 min at 37˚C and fixed with 4% paraformaldehyde for 20 min at room temperature. After washing 3 times with PBS for 5 min, samples were blocked by 10% normal goat serum in PBS for 1hr at room temperature. Next, samples were stained with primary anti-TRPM8 (1:100; #ACC-049, Alomone) and CD41a PerCP-Cy™5.5 (1:100, #340930, BD, Franklin Lakes, NJ) antibodies in PBS with 10% normal goat serum for 1hr in the dark and a Hoechst nuclear stain at 1 ug/ml, washed with PBS three times for 5 min and incubated with secondary antibodies for 1 hr (Goat anti-rabbit-Alexa 647 #21245, Thermo-Fisher, at 1:200). Finally, coverslips were mounted on the 1 mm thick glass slides by incubating with fluoromount (#100241–874, SouthernBiotech) overnight. Cells were visualized using a LEICA SP8X confocal microscope with a 63x objective and tunable White Light Laser system. The data was analyzed using Leica LASX Expert and FIJI software.

Washed platelets from healthy donors were seeded onto poly-Lysine (molecular weight $\geq 300{,}000$ #P1524, Sigma) treated coverslips and permitted to bind for 10 minutes. Coverslips were gently dipped into PBS (1X without calcium or magnesium, Life Technologies) to remove any unbound platelets and are then incubated in Tyrode's buffer (10 mm HEPES (Fisher Scientific), 138 mM NaCl (JT Baker), 5.5 mM glucose (ACROS Organics), 12 mM NaHCO3 (Sigma), 0.36 mM Na2HPO4 (Sigma), 2.9 mM KCl (VWR), 0.4 mM MgCl2 (Fisher Scientific), 0.8 mM CaCl2 (VWR), pH = 7.5, filtered with 0.22 μm filter) at RT for an additional 30 minutes to allow time for platelet spreading. Next, platelets were fixed with 4% paraformaldehyde, blocked with 10% goat serum (Gibco) for 1 hour, and stained. Platelets were not permeabilized. Between staining steps, the substrates were washed in PBS for 5 minutes, 3 times (15 minutes total). Anti-TRMP8 primary antibody (#ACC-049, Alomone) was diluted 1:100 in 10% goat serum and incubated for 1 hour. Goat anti-rabbit-Alexa 647 (#21245, Thermo-Fisher) secondary antibody was diluted 1:200 in 10% goat serum and incubated for 1 hour. Plasma membrane dye R18 (#O246, ThermoFisher) was diluted 1:1000 (to 1 μM) 1:200 in 10% goat serum and incubated for 10 minutes. Substrates were mounted using Fluoromount-G (#100241–874, SouthernBiotech) and imaged with a 60X oil objective (NA = 1.4) on a Nikon A1 Confocal microscope equipped with 488, 561, and 638 nm lasers. Data were analyzed using FiJi and MATLAB software. Regions of interest (ROI) identifying cell boundaries were determined in MATLAB by thresholding and shape-filling the R18 channel staining. Next, average pixel intensity within the ROI, as well as outside of the ROI (background) was measured for the TRPM8 channel. To identify TRPM8 positive cells, we set the cutoff threshold for the intensity to be more than 2x standard deviations of the background signal.

## Platelet aggregation

Washed platelets were re-suspended at a density of $3 \cdot 10^5$ platelets/μL in MTB-2 Tyrode's buffer containing FAF-HSA 0.1% (w/v), apyrase (0.02 U/mL) but no added calcium. Platelet

samples were placed in the aggregometer cuvettes, and 1% fibrinogen and 1 mM $CaCl_2$ were added. Aggregation dose-response curves were generated by the addition of the coagulation agent such as ADP (0.5, 1, 2, 4 μM), Convulxin A (1-10ng/ml) or collagen (0.25–1 ug/ml), and the concentration which provides around 10–20% aggregation (but no more than 60%) was selected for further testing. Before the start of the next recording samples were incubated with vehicle DMSO or PF 05105679 (2 μM) for 5 min. Next, platelets were incubated for 5 min with TRPM8 agonists or corresponding solvent vehicles. Stocks of Menthol and WS-12 were made in ethanol, while icilin and PF 05105679 were made in DMSO. Finally, platelets were subjected to the chosen sub-threshold aggregation-inducing coagulation agent concentration. Data were analyzed using MS Office Excel and GraphPad Prizm.

## Measurement of platelet intracellular calcium

Washed platelets were prepared as described above with slight modifications to accommodate for calcium-sensitive dye loading. In particular, after the first fast centrifugation step platelets were incubated with 3 μM Fura-2 AM (#F1221, Invitrogen) and 0.2% Pluronic F-127 (#P3000MP, Invitrogen) in MTB-1 for 30 min with occasional gentle mixing in the dark. Then, an extra 0.5 μM $PGI_2$ was added and platelets were incubated for an additional 15 min. Next, ACD-A at 1:10 was added immediately prior to a second fast spin at 900g for 5 min. The pellet was re-suspended in MTB-2. Platelet density was adjusted to 300K platelets/μL and they were incubated for additional 30 minutes to allow for de-esterification of Fura-2-AM. Finally, $CaCl_2$ was added at 2 mM. For platelets re-suspended without extracellular calcium all residual Ca2+ was buffered by the addition of 100 μM EGTA). Intracellular calcium levels of a platelet population were recorded using a SpectraMax M5 (Molecular Devices, San Jose, CA) plate reader with 96 well plate adapter. 100 μL of platelets in suspension were loaded in the wells of 96 well plate with UV-permeant glass flat-bottom and black walls (#655096, Greiner Bio-One). Recording temperatures were the same as the temperature during blood collection. Fluorescent images of Fura-2-AM were taken at 340nm and 380nm excitation and 510nm emission with 16 seconds intervals through the bottom of the plate, with a cut-off at 495nm and no mixing between reads. Experimental solutions, such as agonists or their corresponding vehicles, were added into each well by manually pipetting 2 μl of a 50x stock solution and manual stirring. Data was analyzed using MS Office Excel and GraphPad Prizm. When ratiometric dye Fura 2 was used, the actual calcium concentration was calculated according to [57]. When the non-ratiometric dye Fluo 4 was used, fluorescence (F) was normalized to baseline fluorescence ($F_0$).

## Measurement of HEK293T/17 and platelet intracellular calcium during chilling

Washed platelets and HEK293T/17 cells 24 to 48 hours after transfection were prepared as described above with slight modifications to accommodate for calcium-sensitive dye loading. Platelets or HEK293T/17 cells 24 to 48 hours after transfection were loaded with 4 μM Calcium Green™-1, AM (#C3011MP, ThermoFisher) and 0.2% Pluronic F-127 in MTB-1 for 45 minutes at room temperature in the dark. Excess dye was removed by spinning at 800g for 5 minutes and re-suspending in buffer without calcium: MTB-2 –for platelets or HBR–for HEK293T/17 cells. Cells were loaded into 96-well PCR plates (#MSP9601, Bio-Rad) and sealed (#MSB 1001, Bio-Rad). The temperature of the samples was controlled by CFX Connect Real-Time PCR Detection System using CFX Manager Software. The experimental protocol went as follows: holding at 37˚C for 10 minutes; starting at 37˚C until 10˚C temperature steps of -1˚C every 5 seconds; warming back to 37˚C for 10 minutes, the addition of calcium ionophore A23187

(7 μM) for 10 minutes. Calcium Green™-1, AM fluorescence was obtained with excitation at 450-490nm and detection at 515-530nm, measured every 12 seconds. Experimental solutions, such as agonists or their corresponding vehicles, were added into each well by manually pipetting 3 μl of a 10x stock solution and manual stirring at the beginning of the recording. Data was analyzed using MS Office Excel and GraphPad Prism. Change in Calcium Green™-1 fluorescence ($\Delta F$) was calculated by subtracting the baseline ($F_0$) and normalizing to the maximum obtained after the addition of calcium ionophore 7 μM A23187 ($F_{max}$).

## Supporting information

**S1 Fig. TRPM8 gene expression in megakaryocytic lineage during normal hematopoiesis.** Data were obtained from BloodSpot, a gene-centric database of mRNA expression of hematopoietic cells (Bagger et al., 2018). RNAseq was performed by Novershtern et al., 2011, source: GSE24759. The HSCs were identified as CD133-positive and CD34-dim (n = 10), while Megakaryocytes as CD34+, CD41+, CD61+, and CD45-negative (n = 7). Error bars indicate Mean ± SEM. Statistical analysis was performed using an unpaired Student t-test, where asterisks indicated a p = 0.009.
(TIF)

**S2 Fig. RBC- and WBC-depleted platelet preparation shows CD45 and TRPM8 signals.** Agarose gel electrophoresis of PCR products from CD45-and CD235a (Glycophorin A)- depleted platelet preparation. PRP was depleted of CD45-positive cells using magnetic microbeads using AutoMACS sorter. **A**. PCR reaction using CD45 primers from CD45-positive cells, platelet preparations from two separate donors, and mock. **B**. PCR reaction using the TRPM8 1410F/1788R primer set. Arrows indicate the size of the expected amplicons: 300bp for the CD45 primer set; and 379 bp for the 1410F/1788R primer set.
(TIF)

**S3 Fig. TRPM8 receptor protein in human platelets by immunoblotting. A**. Western blot from HEK293T/17 cell lysates, transfected with GFP or TRPM8-GFP. Anti-TRPM8 (ACC-049) was used with or without blocking peptide (BLP-CC049). The expected size for TRPM8-GFP fusion protein is ~160kDa (two ways arrow). Blocking peptide subtracted line scan (BP sub line scan) was calculated by measuring pixel intensity along a line drawn down the lanes, normalizing to a background at high molecular weight, and subtracting the values measured for the corresponding lanes with blocking peptide (blue for TRPM8-GFP lane; orange for GFP). Arrow in the line scan indicates a full-length TRPM8-GFP protein. **B**. Western blot of washed platelet lysates from three healthy donors. Line scan was calculated as in **A**. Arrows indicate potential TRPM8 protein. **C**. Representative images of random TRPM8-positive platelets population and CD45 (-/+) staining from one healthy donor by imaging flow cytometry. 20,000 events were measured for each sample. The scale bar is 7 μm.
(TIF)

**S4 Fig. (A-C)** TRPM8 expression in platelets, WBC-derived microparticles, and platelets incubated with WBC-derived microparticles. Platelet-rich plasma was incubated with WBC-derived microparticles for 15 min at RT. Platelet-rich plasma alone, WBC-derived microparticles alone, and platelet-rich plasma incubated with WBC-derived microparticles were stained with TRPM8 antibody with a secondary FITC-labeled antibody, and the pan WBC CD45 antibody, along with CD61 and read by conventional flow cytometry **(A)** and imaging flow cytometry **(B-C)**. **B**. Representative CD45 positive platelets (CD61+ and FSC/SSC gated). **C**. Representative CD45 negative platelets (CD61+ and FSC/SSC gated). Data in (A) are shown as individual data points, mean ± standard error of the mean. N = 3 independent experiments.

**p = 0.0029 for PLT versus PLT+WBC-MP, and **p = 0.0037 for WBC-MP versus PLT+-WBC-MP. Ns = not significant (p = 0.88). Statistical analysis was performed using an One Way ANOVA with Tukey correction for multiple comparisons and assumed equal sphericity. White bar indicates 7μm.
(TIF)

**S5 Fig. (A-B)** Change in Calcium Green™-1 fluorescence levels baseline subtracted and normalized to maximum obtained after addition of calcium ionophore 7 μM A23187. HEK293T/17 cells transfected with TRPM8 (**A**) or empty vector (**B**), untransfected) were suspended in HEPES buffered saline containing either 0 mM $Ca^{2+}$ and 100 μM EGTA (black) with vehicle DMSO, 2 mM $Ca^{2+}$ with vehicle DMSO (gray) or 2 mM $Ca^{2+}$ with 2 μM PF 05105679 (blue). **C.** Quantification of maximal calcium increase at 10˚C in HEK293T/17 cells, n = 2. **D.** The overlay of the linear fit (dashed line, $R^2$ = 0.82) of the average negative control—the un-transfected HEK cells in 2 mM $Ca^{2+}$ (same as in B, gray) and the average calcium response in washed platelets in 0 mM $Ca^{2+}$ and 100 μM EGTA -containing Tyrode's buffer with vehicle DMSO (gray, same as Fig 7C) or 5 μM thapsigargin (pink, same as Fig 7C). Arrow indicates an apparent threshold for platelet activation at ~ 23˚C.
(TIF)

**S6 Fig. TRPM8 agonists do not lead to activation of the human washed platelets after 1 hour of incubation. (A-B)** Human washed platelets were evaluated via flow cytometry. **A**. Integrin αIIbβ3 activation in samples stained with PAC-1 fluorescent antibody (MFI normalized to vehicle). **B**. Alpha granule release as seen from P-selectin externalization (anti-P-selectin fluorescent antibody MFI normalized to vehicle). Initially, samples were pre-incubated with either vehicle DMSO or PF 05105679 (2 μM) for 5 minutes. Next, samples were treated with either vehicle (Ethanol), menthol (500 μM), WS-12 (2 μM) or icilin (100 μM) for 1 hour at either 22˚C (white background) or 4˚C temperature (green background). Values were normalized to those measured in platelets treated with the vehicle at 22˚C. (**C-E**) Samples were evaluated via imaging flow cytometry. **C**. Percent microaggregates in samples treated the same as in A and B. (**D, E**) Percent spheroid (**D**) or discoid (**E**) cells in samples treated same as in A and B. Lines connecting data points indicate the same donor. Statistical analysis was performed using paired Student t-test, where asterisks indicated a p-value lower than 0.05 for *, and "ns" indicates s p-value >0.05. Symbols above brackets indicate paired comparison between treatment groups, and without bars indicate comparison to vehicle.
(TIF)

**S7 Fig. TRPM8 agonists do not lead to activation of the human washed platelets after 4 hours of incubation. (A-B)** Human washed platelets were evaluated via flow cytometry. **A**. Integrin αIIbβ3 activation in samples stained with PAC-1 fluorescent antibody (MFI normalized to vehicle). **B**. Alpha granule release as seen from P-selectin externalization (anti-P-selectin fluorescent antibody MFI normalized to vehicle). Initially, samples were pre-incubated with either vehicle DMSO or PF 05105679 (2 μM) for 5 minutes. Next, samples were treated with either vehicle (Ethanol), menthol (500 μM), WS-12 (2 μM) or icilin (100 μM) for 4 hours at either 22˚C (white background) or 4˚C temperature (green background). Values were normalized to those measured in platelets treated with vehicle. (**C-E**) Samples were evaluated via imaging flow cytometry. **C.** Percent microaggregates in samples treated the same as in A and B. (**D, E**) Percent spheroid (**D**) or discoid (**E**) cells in samples treated the same as in A and B. Lines connecting data points indicate the same donor. Statistical analysis was performed using paired Student t-test, where asterisks indicated a p-value lower than 0.05 for *, and "ns" indicates s p-value >0.05. Symbols above brackets indicate paired comparison between treatment

groups, and without bars indicate comparison to vehicle.
(TIF)

**S1 File.**
(DOCX)

**S2 File.**
(XLSX)

**S1 Raw images.**
(PDF)

# Acknowledgments

The authors would like to thank Renetta Stevens and Tena Petersen for their administrative support. Katie Benson (Bloodworks Bio, Seattle, WA) helped us by isolating CD45-negative platelets for PCR. Jill Jensen (University of Washington, Seattle, WA) for kindly sharing a homemade perfusion chamber. Sharona E. Gordon (University of Washington, Seattle, WA) for extensive discussions of the project.

# Author Contributions

**Conceptualization:** José A. López, Moritz Stolla.

**Data curation:** Anastasiia Stratiievska, Olga Filippova, Tahsin Özpolat, Daire Byrne, S. Lawrence Bailey, Aastha Chauhan, Molly Y. Mollica, Jeff Harris, Kali Esancy, Junmei Chen.

**Formal analysis:** Anastasiia Stratiievska, Olga Filippova, Tahsin Özpolat, Daire Byrne, S. Lawrence Bailey, Molly Y. Mollica, Jeff Harris, Kali Esancy, Junmei Chen, Ajay K. Dhaka, Nathan J. Sniadecki.

**Funding acquisition:** José A. López, Moritz Stolla.

**Investigation:** Anastasiia Stratiievska, Olga Filippova, Tahsin Özpolat, S. Lawrence Bailey, Aastha Chauhan, Junmei Chen, Ajay K. Dhaka, Nathan J. Sniadecki, José A. López, Moritz Stolla.

**Methodology:** Anastasiia Stratiievska, Tahsin Özpolat, Junmei Chen, Ajay K. Dhaka, Nathan J. Sniadecki.

**Project administration:** Moritz Stolla.

**Supervision:** Junmei Chen, Ajay K. Dhaka, Nathan J. Sniadecki, José A. López, Moritz Stolla.

**Validation:** Anastasiia Stratiievska, Junmei Chen.

**Visualization:** Anastasiia Stratiievska.

**Writing – original draft:** Anastasiia Stratiievska.

**Writing – review & editing:** Olga Filippova, Tahsin Özpolat, Daire Byrne, S. Lawrence Bailey, Aastha Chauhan, Molly Y. Mollica, Jeff Harris, Kali Esancy, Junmei Chen, Ajay K. Dhaka, Nathan J. Sniadecki, José A. López, Moritz Stolla.

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
