## [Decision Letter · Decision Letter 0]

28 Sep 2023

PONE-D-23-22002Cold temperature induces a TRPM8-independent calcium release from the endoplasmic reticulum in human plateletsPLOS ONE

Dear Dr. Stolla,

Thank you for submitting your manuscript to PLOS ONE. After careful consideration, we feel that it has merit but does not fully meet PLOS ONE’s publication criteria as it currently stands. Therefore, we invite you to submit a revised version of the manuscript that addresses the points raised during the review process.

We look forward to receiving your revised manuscript.

Kind regards,

Naim Akhtar Khan, PhD, DSc

Academic Editor

PLOS ONE

“Funding sources from NIH: HL147462 and HL007093 to MYM; S10 OD016240 to Keck center; 5R01NS115747-03 to AKD; HL145262 and HL149734 to NJS; R35HL145262 JAL; 1R01HL153072-01 to MS; and institutional funds from the Bloodworks Northwest.”

“M.S. received research funding from Terumo BCT and Cerus Corp. All other authors have no COI to declare.”

“The authors would like to thank Renetta Stevens and Tena Petersen for their administrative support. Katie Benson (Bloodworks Bio, Seattle, WA) help us for isolating CD45-negative platelets for PCR. Jill Jensen (University of Washington, Seattle, WA) for kindly sharing a homemade perfusion chamber. Sharona E. Gordon (University of Washington, Seattle, WA) for extensive discussions of the project. Funding sources from NIH: HL147462 and HL007093 to MYM; S10 OD016240 to Keck center; 5R01NS115747-03 to AKD; HL145262 and HL149734 to NJS; R35HL145262 JAL; 1R01HL153072-01 to MS; and institutional funds from the Bloodworks Northwest.”

“Funding sources from NIH: HL147462 and HL007093 to MYM; S10 OD016240 to Keck center; 5R01NS115747-03 to AKD; HL145262 and HL149734 to NJS; R35HL145262 JAL; 1R01HL153072-01 to MS; and institutional funds from the Bloodworks Northwest.”

6. We note that you have indicated that data from this study are available upon request. PLOS only allows data to be available upon request if there are legal or ethical restrictions on sharing data publicly. For more information on unacceptable data access restrictions, please see http://journals.plos.org/plosone/s/data-availability#loc-unacceptable-data-access-restrictions.

7. We note that you have included the phrase “data not shown” in your manuscript. Unfortunately, this does not meet our data sharing requirements. PLOS does not permit references to inaccessible data. We require that authors provide all relevant data within the paper, Supporting Information files, or in an acceptable, public repository. Please add a citation to support this phrase or upload the data that corresponds with these findings to a stable repository (such as Figshare or Dryad) and provide and URLs, DOIs, or accession numbers that may be used to access these data. Or, if the data are not a core part of the research being presented in your study, we ask that you remove the phrase that refers to these data.

8. Please include your full ethics statement in the ‘Methods’ section of your manuscript file. In your statement, please include the full name of the IRB or ethics committee who approved or waived your study, as well as whether or not you obtained informed written or verbal consent. If consent was waived for your study, please include this information in your statement as well.

9. PLOS ONE now requires that authors provide the original uncropped and unadjusted images underlying all blot or gel results reported in a submission’s figures or Supporting Information files. This policy and the journal’s other requirements for blot/gel reporting and figure preparation are described in detail at https://journals.plos.org/plosone/s/figures#loc-blot-and-gel-reporting-requirements and https://journals.plos.org/plosone/s/figures#loc-preparing-figures-from-image-files. When you submit your revised manuscript, please ensure that your figures adhere fully to these guidelines and provide the original underlying images for all blot or gel data reported in your submission. See the following link for instructions on providing the original image data: https://journals.plos.org/plosone/s/figures#loc-original-images-for-blots-and-gels.

10. We notice that your supplementary file included in the manuscript file. Please remove them and upload them with the file type 'Supporting Information'. Please ensure that each Supporting Information file has a legend listed in the manuscript after the references list.

11. We notice that your supplementary figures are uploaded with the file type 'Figure'. Please amend the file type to 'Supporting Information'. Please ensure that each Supporting Information file has a legend listed in the manuscript after the references list.

Additional Editor Comments:

The MS is interesting, however, it requires substantial revision that will involve the complementary experiments to prove some hypotheses as suggested by the second reviewer.

Reviewers' comments:

Reviewer's Responses to Questions

**Comments to the Author**

1. Is the manuscript technically sound, and do the data support the conclusions?

Reviewer #1: Yes

Reviewer #2: Partly

2. Has the statistical analysis been performed appropriately and rigorously? 

Reviewer #1: Yes

Reviewer #2: Yes

3. Have the authors made all data underlying the findings in their manuscript fully available?

Reviewer #1: Yes

Reviewer #2: Yes

4. Is the manuscript presented in an intelligible fashion and written in standard English?

Reviewer #1: Yes

Reviewer #2: Yes

5. Review Comments to the Author

Reviewer #1: This study investigates the role of TRPM8 in platelets and they conclude that TRPM8 contributes to some features of the acute cold response in platelets but is dispensable for acute cold-induced calcium increase. This manuscript is well-written and well-designed with significant scientific novelty. I only have one small question.

1. Figure 7 shows that chilling platelets leads to a rapid calcium increase independent of TRPM8 using TRPM8 inhibitor PF 05105679, however, in Figure 2 they concluse that a small proportion of human platelets express TRPM8, which might lead to one question. Is the reason why TRPM8 is ineffective in human platelets due to low expression of TRPM8, or is it because TRPM8 is not primarily expressed in the cell membrane in human platelets? Or is it that TRPM8 inhibitors cannot act on the TRPM8 channel of human platelets? I think the cell membrane of human platelets should be separated from ER to detect the expression and localization of TRPM8. Perhaps TRPM8 is also present on ER, and TRPM8 inhibitors cannot enter the cells, resulting in the illusion that the rapid calcium increase in chilling platelets did not depend on TRPM8.

Reviewer #2: This manuscript describes presence of TRMP8 channel protein in a human megakaryocyte cell line (MEG1) and in platelets obtained from healthy humans. The MEG1 cells contain the TRMP8 mRNA and the protein, which is uniformly distributed on the membrane. Human platelets are positive for mRNA but there is concern this could be due to contamination of the platelet preparation with leukocytes which also express TRMP8. There is evidence of the TRMP8 protein being present on platelets but only in few spots, which could be due to microparticles from other cells (leukocytes) attached to platelets. The manuscript goes on to describe multiple experiments aimed at defining a functional role of the TRMP8 protein in platelets. However, majority of these experiments are negative for demonstration of any functionality, including data related to a cationic channel function of the protein and calcium influx. The only hint of function is related to PAC-1 binding induced by changing the temperature of the platelets. There is low PAC-1 binding at 37 C, which increases at 22C but decreases at 4 C. Thus, even this data is not consistent.

The presence of TRMP8 in platelet preparations could be due to contamination with microparticles from other cells, as is mentioned by the authors in the discussion. Supplementary Figure 3 addresses this concern to some degree, but the microscope images are from single cells. Additional experiments should be done to eliminate this possibility, including incubation of TRMP8 negative platelets with plasma that contains microparticles from CD45 + cells. A stronger argument would be to use flow cytometry detection of TRMP8 (+) platelets after TRMP8(-) platelets were incubated with plasma which contains microparticles positive for TRMP8. Such an experiment would show if platelets can pick up microparticles with the TRMP8 protein.

The effect of cold temperature increasing intracellular calcium levels in platelets has been known for a long time. The new aspect of this study would be involvement of TRMP8 in this process, but it does not seem to be the case based on the data presented. Thus, even though this manuscript is full of elegant and impressive experiments, these are for most part with negative results and overall do not add to the body of knowledge for human platelets. The MEG1 cell line appears to have more uniform distribution of the TRMP8 protein and thus it may be worthwhile to study the function of TRMP8 protein in these cells.

Minor points

Figure 5, Panel D. Collagen is misspelled.

Figure 7, legend for Panel D, should have D instead of G.

6. PLOS authors have the option to publish the peer review history of their article (what does this mean?). If published, this will include your full peer review and any attached files.

Reviewer #1: **Yes: **Peng Gao

Reviewer #2: No

---

## [Author Response · Author response to Decision Letter 0]

26 Jan 2024

We would like to thank the editor and the reviewers for their positive evaluation of our manuscript and their insightful feedback. We are grateful for the suggested changes because they have improved this manuscript significantly. Please find our point-by-point responses below.

We double-checked the requirements and made the appropriate corrections.

We corrected this in the current version and added the funding from the CoI statement.

“Funding sources from NIH: HL147462 and HL007093 to MYM; S10 OD016240 to Keck center; 5R01NS115747-03 to AKD; HL145262 and HL149734 to NJS; R35HL145262 JAL; 1R01HL153072-01 to MS; and institutional funds from the Bloodworks Northwest.” M.S. received research funding from Terumo BCT and Cerus Corp. All other authors have no COI to declare.

We added this statement to the revised cover letter.

“M.S. received research funding from Terumo BCT and Cerus Corp. All other authors have no COI to declare.”

We added this statement to the updated cover letter.

“The authors would like to thank Renetta Stevens and Tena Petersen for their administrative support. Katie Benson (Bloodworks Bio, Seattle, WA) help us for isolating CD45-negative platelets for PCR. Jill Jensen (University of Washington, Seattle, WA) for kindly sharing a homemade perfusion chamber. Sharona E. Gordon (University of Washington, Seattle, WA) for extensive discussions of the project. Funding sources from NIH: HL147462 and HL007093 to MYM; S10 OD016240 to Keck center; 5R01NS115747-03 to AKD; HL145262 and HL149734 to NJS; R35HL145262 JAL; 1R01HL153072-01 to MS; and institutional funds from the Bloodworks Northwest.”

“Funding sources from NIH: HL147462 and HL007093 to MYM; S10 OD016240 to Keck center; 5R01NS115747-03 to AKD; HL145262 and HL149734 to NJS; R35HL145262 JAL; 1R01HL153072-01 to MS; and institutional funds from the Bloodworks Northwest.”

We removed our funding statement from the manuscript and added an updated one to the revised cover letter.

6. We note that you have indicated that data from this study are available upon request. PLOS only allows data to be available upon request if there are legal or ethical restrictions on sharing data publicly. For more information on unacceptable data access restrictions, please see http://journals.plos.org/plosone/s/data-availability#loc-unacceptable-data-access-restrictions.

There are no restrictions. We added the data set to the revised submission.

7. We note that you have included the phrase “data not shown” in your manuscript. Unfortunately, this does not meet our data sharing requirements. PLOS does not permit references to inaccessible data. We require that authors provide all relevant data within the paper, Supporting Information files, or in an acceptable, public repository. Please add a citation to support this phrase or upload the data that corresponds with these findings to a stable repository (such as Figshare or Dryad) and provide and URLs, DOIs, or accession numbers that may be used to access these data. Or, if the data are not a core part of the research being presented in your study, we ask that you remove the phrase that refers to these data.

We removed this sentence because it does not add any critical value to the manuscript.

8. Please include your full ethics statement in the ‘Methods’ section of your manuscript file. In your statement, please include the full name of the IRB or ethics committee who approved or waived your study, as well as whether or not you obtained informed written or verbal consent. If consent was waived for your study, please include this information in your statement as well.

We checked our manuscript but we could not find any violation of the requested guidelines. We updated the IRB name from WIRB to WCG-IRB and changed the header from “Healthy human research” to “Ethics statement”

9. PLOS ONE now requires that authors provide the original uncropped and unadjusted images underlying all blot or gel results reported in a submission’s figures or Supporting Information files. This policy and the journal’s other requirements for blot/gel reporting and figure preparation are described in detail at https://journals.plos.org/plosone/s/figures#loc-blot-and-gel-reporting-requirements and https://journals.plos.org/plosone/s/figures#loc-preparing-figures-from-image-files. When you submit your revised manuscript, please ensure that your figures adhere fully to these guidelines and provide the original underlying images for all blot or gel data reported in your submission. See the following link for instructions on providing the original image data: https://journals.plos.org/plosone/s/figures#loc-original-images-for-blots-and-gels.

We added all complete images to our resubmission.

We added this information to the cover letter.

10. We notice that your supplementary file included in the manuscript file. Please remove them and upload them with the file type 'Supporting Information'. Please ensure that each Supporting Information file has a legend listed in the manuscript after the references list.

We removed this information from the main manuscript and added it to the Supporting Information file. 

11. We notice that your supplementary figures are uploaded with the file type 'Figure'. Please amend the file type to 'Supporting Information'. Please ensure that each Supporting Information file has a legend listed in the manuscript after the references list.

We changed the file assignment and added the Supplemental Figure legends to the main manuscript after the references.

Additional Editor Comments:

The MS is interesting, however, it requires substantial revision that will involve the complementary experiments to prove some hypotheses as suggested by the second reviewer.

Reviewers' comments:

Reviewer's Responses to Questions

Comments to the Author

1. Is the manuscript technically sound, and do the data support the conclusions?

Reviewer #1: Yes

Reviewer #2: Partly

2. Has the statistical analysis been performed appropriately and rigorously? 

Reviewer #1: Yes

Reviewer #2: Yes

3. Have the authors made all data underlying the findings in their manuscript fully available?

Reviewer #1: Yes

Reviewer #2: Yes

4. Is the manuscript presented in an intelligible fashion and written in standard English?

Reviewer #1: Yes

Reviewer #2: Yes

5. Review Comments to the Author

Reviewer #1: This study investigates the role of TRPM8 in platelets and they conclude that TRPM8 contributes to some features of the acute cold response in platelets but is dispensable for acute cold-induced calcium increase. This manuscript is well-written and well-designed with significant scientific novelty. I only have one small question.

1. Figure 7 shows that chilling platelets leads to a rapid calcium increase independent of TRPM8 using TRPM8 inhibitor PF 05105679, however, in Figure 2 they concluse that a small proportion of human platelets express TRPM8, which might lead to one question. Is the reason why TRPM8 is ineffective in human platelets due to low expression of TRPM8, or is it because TRPM8 is not primarily expressed in the cell membrane in human platelets? Or is it that TRPM8 inhibitors cannot act on the TRPM8 channel of human platelets? I think the cell membrane of human platelets should be separated from ER to detect the expression and localization of TRPM8. Perhaps TRPM8 is also present on ER, and TRPM8 inhibitors cannot enter the cells, resulting in the illusion that the rapid calcium increase in chilling platelets did not depend on TRPM8.

We thank this reviewer for this insightful comment. Given the pharmacologic properties and structure of the small molecule inhibitor (PF 05105679) we used, there is little doubt that this inhibitor acts intracellularly. In their publication, Andrews et al. describe PF 05105679’s ability to permeate, overall lipid solubility, and moderate polarity in their publication (ACS Med. Chem. Lett. 2015, 6, 419−424).

Reviewer #2: This manuscript describes presence of TRMP8 channel protein in a human megakaryocyte cell line (MEG1) and in platelets obtained from healthy humans. The MEG1 cells contain the TRMP8 mRNA and the protein, which is uniformly distributed on the membrane. Human platelets are positive for mRNA but there is concern this could be due to contamination of the platelet preparation with leukocytes which also express TRMP8. There is evidence of the TRMP8 protein being present on platelets but only in few spots, which could be due to microparticles from other cells (leukocytes) attached to platelets. The manuscript goes on to describe multiple experiments aimed at defining a functional role of the TRMP8 protein in platelets. However, majority of these experiments are negative for demonstration of any functionality, including data related to a cationic channel function of the protein and calcium influx. The only hint of function is related to PAC-1 binding induced by changing the temperature of the platelets. There is low PAC-1 binding at 37 C, which increases at 22C but decreases at 4 C. Thus, even this data is not consistent.

The presence of TRMP8 in platelet preparations could be due to contamination with microparticles from other cells, as is mentioned by the authors in the discussion. Supplementary Figure 3 addresses this concern to some degree, but the microscope images are from single cells. Additional experiments should be done to eliminate this possibility, including incubation of TRMP8 negative platelets with plasma that contains microparticles from CD45 + cells. A stronger argument would be to use flow cytometry detection of TRMP8 (+) platelets after TRMP8(-) platelets were incubated with plasma which contains microparticles positive for TRMP8. Such an experiment would show if platelets can pick up microparticles with the TRMP8 protein.

We appreciate this reviewer’s feedback and helpful suggestions. We found TRPM8 expression overall too low to allow for successful sorting between TRPM8(+) and TRPM8(-) events by flow cytometry. Nevertheless, we generated microparticles from white blood cells with the addition of an ionophore. We then separated these microparticles from larger WBC cell remnants and incubated them with platelets. TRPM8 expression was then tested with and without WBC-MP incubation by conventional and imaging flow cytometry. We found a significant increase in TRPM8 expression incubation with WBC-MP proving the general ability of WBC-MP to transfer TRPM8 to platelets. However, TRPM8 expression on platelets was not dependent on WBC-MP, as WBC- MP-negative platelets also expressed TRPM8, albeit to a lesser extent. 

The effect of cold temperature increasing intracellular calcium levels in platelets has been known for a long time. The new aspect of this study would be involvement of TRMP8 in this process, but it does not seem to be the case based on the data presented. Thus, even though this manuscript is full of elegant and impressive experiments, these are for most part with negative results and overall do not add to the body of knowledge for human platelets. The MEG1 cell line appears to have more uniform distribution of the TRMP8 protein and thus it may be worthwhile to study the function of TRMP8 protein in these cells.

We agree with the assessment of this reviewer. Even though most of our data are negative, we still consider it worth reporting to avoid that other investigators make similar efforts. In addition, we provide novel evidence that cold temperature induces ER-dependent Ca2+ release.

Minor points

Figure 5, Panel D. Collagen is misspelled.

We thank the reviewer for catching this error. We changed this accordingly. 

Figure 7, legend for Panel D, should have D instead of G.

We thank the reviewer for catching this error. We changed this accordingly. 

6. PLOS authors have the option to publish the peer review history of their article (what does this mean?). If published, this will include your full peer review and any attached files.

Do you want your identity to be public for this peer review? For information about this choice, including consent withdrawal, please see our Privacy Policy.

Reviewer #1: Yes: Peng Gao

Reviewer #2: No

---

## [Editor Report · Decision Letter 1]

31 Jan 2024

Cold temperature induces a TRPM8-independent calcium release from the endoplasmic reticulum in human platelets

PONE-D-23-22002R1

Dear Dr. Stolla,

We’re pleased to inform you that your manuscript has been judged scientifically suitable for publication and will be formally accepted for publication once it meets all outstanding technical requirements.

Kind regards,

Naim Akhtar Khan, PhD, DSc

Academic Editor

PLOS ONE
---

## [Editor Report · Acceptance letter]

22 Feb 2024

PONE-D-23-22002R1 

PLOS ONE

Dear Dr. Stolla, 

I'm pleased to inform you that your manuscript has been deemed suitable for publication in PLOS ONE. Congratulations! Your manuscript is now being handed over to our production team.

Kind regards, 

on behalf of

Professor Naim Akhtar Khan 

Academic Editor

PLOS ONE